# `NewTerm`: Benchmarking Real-Time New Terms for Large Language Models with Annual Updates

**Hexuan Deng**[1]    **Wenxiang Jiao**    **Xuebo Liu**[1][*]    **Min Zhang**[1]    **Zhaopeng Tu**

[1] Institute of Computing and Intelligence, Harbin Institute of Technology, Shenzhen, China

{hxuandeng,wenxiangjiaonju,tuzhaopeng}@gmail.com,
{liuxuebo,zhangmin2021}@hit.edu.cn

## Abstract

Despite their remarkable abilities in various tasks, large language models (LLMs) still struggle with real-time information (e.g., new facts and terms) due to the knowledge cutoff in their development process. However, existing benchmarks focus on outdated content and limited fields, facing difficulties in real-time updating and leaving new terms unexplored. To address this problem, we propose an adaptive benchmark, NewTerm, for real-time evaluation of new terms. We design a highly automated construction method to ensure high-quality benchmark construction with minimal human effort, allowing flexible updates for real-time information. Empirical results on various LLMs demonstrate over 20% performance reduction caused by new terms. Additionally, while updates to the knowledge cutoff of LLMs can cover some of the new terms, they are unable to generalize to more distant new terms. We also analyze which types of terms are more challenging and why LLMs struggle with new terms, paving the way for future research. Finally, we construct NewTerm 2022 and 2023 to evaluate the new terms updated each year and will continue updating annually. The benchmark and codes can be found at https://github.com/hexuandeng/NewTerm.

## 1   Introduction

Large language models (LLMs) have shown remarkable progress, achieving impressive performance on various benchmarks across multiple domains [7, 9, 23, 37, 38]. However, they struggle with real-time interaction [44, 70], which is crucial and challenging. In the constantly evolving internet landscape, real-time information like new facts and terms continuously emerge, where LLMs are expected to perform well.

Recent work has designed benchmarks to evaluate the performance of LLMs on new facts and the effectiveness of various improvement methods [14, 46, 53, 78]. However, benchmarks based on new terms have not been well-studied yet, which is a crucial problem that significantly reduces model performance [43, 49]. We urgently need a real-time benchmark to annually evaluate the performance of different LLMs and potential improvement methods toward new terms.

Besides, as the knowledge cutoff of LLMs is constantly updated, benchmarks for real-time information will soon become outdated. However, most ex-

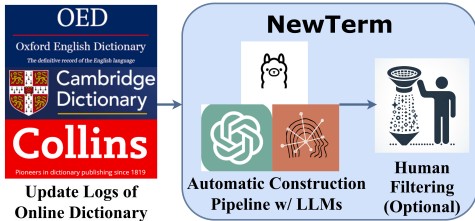

Figure 1: The framework for constructing the benchmark based on real-time new terms from the dictionary.

---

[*] Corresponding Author

38th Conference on Neural Information Processing Systems (NeurIPS 2024) Track on Datasets and Benchmarks.

isting benchmark construction methods heavily rely on human efforts [31, 32, 56, 64, 67], making real-time updates extremely costly. With the rapid development of LLMs, a highly automated construction of real-time benchmarks, which can be continuously updated at a low cost, is invaluable.

To address these issues, we develop a highly automatic construction method for adaptively benchmarking new terms. As illustrated in Figure 1, we first collect new terms from online dictionaries, covering new words, new phrases, and old words with new meanings. Then, we employ LLMs to automatically construct the benchmark. Human filtering reveals that automatic construction has over 80% accuracy. Additionally, pre- and post-filtering evaluation results are highly consistent, indicating our benchmark can effectively evaluate LLMs with no human effort. Finally, we obtain the new term benchmark, NewTerm 2022 and NewTerm 2023, and will continue to update it annually.

Empirical results from over twenty diverse LLMs demonstrate the significant challenge new terms pose, with over 20% accuracy decrease when LLMs do not understand the new terms in the question. Furthermore, we construct detailed analyses over years and new term features, aiming to pave the way for developing more effective approaches toward new terms. Our contributions are:

- We automatically benchmark real-time new terms for LLMs. Empirical results reveal that new terms significantly reduce the performance of LLMs.
- We reveal the trends in performance variation with respect to LLM and new terms changes over years. We find that updates to LLMs' knowledge cutoff often do not encompass all new terms, and the overlap of terms learned by different series of LLMs is limited.
- Further analysis reveals which terms are more difficult based on term type, frequency, and deducing difficulty. We also analyze the reasons LLMs struggle with new terms.
- We publicly release the code, the constructed challenging benchmark, and the evaluation code to facilitate future research. We will release benchmarks annually to evaluate terms from the latest year, thereby tracking the real-time performance of the most recent LLMs.

## 2 Related Work

**Real-time benchmark.** Several benchmarks have been developed to assess the ability of models to acquire new knowledge. Levy et al. [41] and Meng et al. [46] focus on QA tasks that incorporate altered facts, while Mallen et al. [44] targets long-tail questions. Cheang et al. [12] focus on abstractive summarization tasks, and Arodi et al. [4] on coreference resolution tasks, both of which pose significant challenges when models contain outdated knowledge. Kasai et al. [40] introduce a dynamic QA platform that evaluates novel events or information on a regular basis. Yu et al. [73] construct a knowledge-oriented LLM assessment benchmark for world knowledge evaluation. Recently, more benchmarks have been proposed for multi-hop QA tasks. Yin et al. [72] introduces an artificial fact and multi-hop question generation approach, while Zhong et al. [78] focuses on real-world fact updates. Cohen et al. [14] provides broader and finer-grained categories for determining when a fact should change under multi-hop settings or not.

However, few benchmarks are specifically designed for new terms, which we aim to focus on. Martínez et al. [45] directly query LLMs about their knowledge of these terms, resulting in a lack of robustness towards updates of LLMs and different prompts. Recently, Zheng et al. [76] revealed performance degradation in NLG tasks. But it heavily relies on human effort, making updates costly and will be out of date as LLMs continue updating. In contrast, our approach focuses on NLU tasks, and will be updated annually, thanks to our highly automated construction pipeline.

**LLM as data generator.** Significant advancements have been made in generating training data using teacher LLMs [11, 16, 17, 29, 30, 47, 48, 52]. To address the unreliability of LLMs as evaluators, some studies have attempted to use strong LLMs like ChatGPT [50] or GPT-4 [51] to construct benchmarks through well-designed filtering methods. Jain et al. [34] propose a self-supervised evaluation framework for LLMs that monitors behavior on real-world datasets. Qin et al. [54] introduce an automatic evaluator for multi-tool usage evaluation. Yin et al. [72] generate a question-answering (QA) dataset for new fact evaluation based on knowledge chains as triples.

However, while the generating quality of LLMs is ideal, comparable with human annotators [26], the building process is task-dependent. For new terms, the input information is limited, i.e., only the term and its meaning are available, making the construction more complicated.

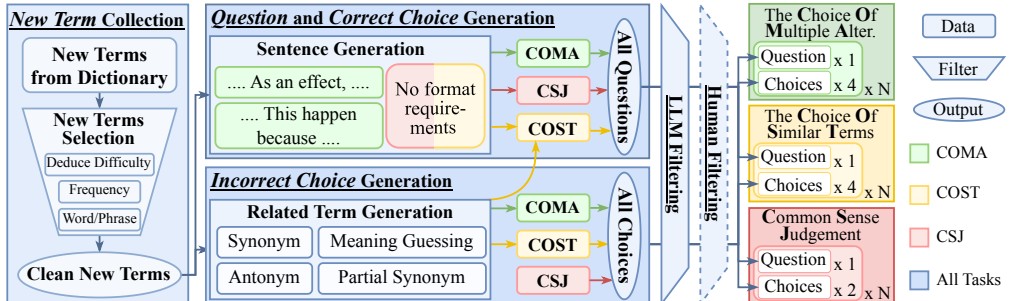

Figure 2: The construction pipeline for NewTerm benchmark. We use different colors to indicate the parts used by each task, with COMA, COST, and CSJ represented by green, yellow, and red.

# 3    Contructing NewTerm Benchmark

We introduce our construction of the new terms benchmark, NewTerm. We first collect new terms from online dictionaries (Section 3.2) and design three open-domain tasks to test the model's understanding of these new terms (Section 3.3). We then detail our benchmark construction pipeline (Section 3.4). Finally, we filter the benchmark to ensure quality and analyze human annotations (Section 3.5).

## 3.1    Design Principle

**Covering diverse tasks and terms.**    To comprehensively evaluate the impact of new terms, we design benchmarks for three distinct open-domain tasks in English, each focusing on a fundamental aspect of the abilities of LLMs. Furthermore, we concentrate on three typical categories of new terms: new words, new phrases, and old words with new meanings.

**Tracking annual updates of LLMs and new terms.**    As the knowledge cutoff of LLMs is continuously updated and new knowledge constantly emerges, tracking the performance of various series of LLMs towards new terms from different time periods is crucial. Therefore, we construct benchmarks on an annual basis, currently covering 2022 and 2023. We will update the benchmark annually, utilizing the highly automatic pipeline and incorporating a broader coverage of new terms. This allows us to analyze LLM performance in terms of both model updates and new term updates.

**Highly automated benchmark construction.**    To enable annual updates of benchmarks, a low-cost, highly automated construction approach is needed. To this end, we carefully design a construction framework that automatically builds benchmarks step by step, allowing LLMs to create high-quality benchmarks. The high consistency with human annotations demonstrates that we can automatically construct and update benchmarks that can effectively evaluate LLMs without any human effort.

**Potential effect.**    Our construction pipeline, as the first highly automated method benchmarking real-time information, can not only track the knowledge updates of LLMs on new term understanding, but also evaluate potential improvement strategies, e.g., model editing [20, 25, 33], test-time adaptation [59, 65, 75], and retrieval [36, 42, 44]. Moreover, to encourage periodic updates for real-time performance benchmarking, we provide construction inspirations and techniques for future work.

## 3.2    New Term Collection

**New terms from dictionary.**    In this study, we focus on terms added to dictionaries each year. We collect these terms from the update logs of three prominent online dictionaries: Cambridge, Collins, and Oxford. Currently, 4.2k terms added in 2022 (January 2022 to March 2023) and 2.9k terms in 2023 (April 2023 to March 2024) are collected, with continuous updates in the future.

**New terms selection.**    We mainly focus on three typical categories of new terms: new words, new phrases, and old words with new meanings, which pose significant challenges to models [43, 49, 76]. As representatives, we select 300 new terms each year, evenly distributed across three categories. To select terms that best fit these categories, we classify the collected terms across several dimensions:

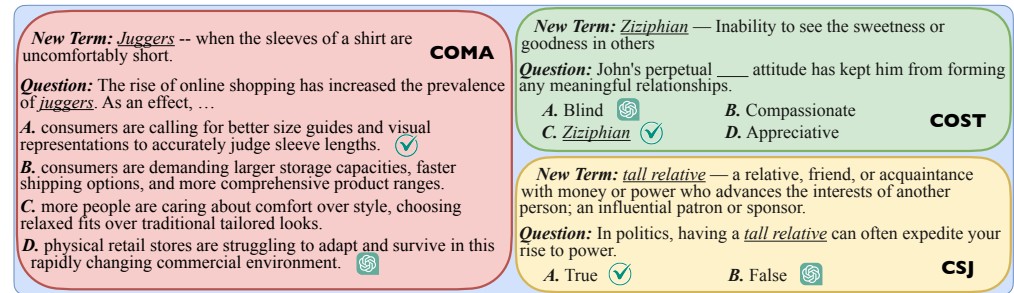

Figure 3: Examples of three open-domain NLU tasks in NewTerm benchmark. The choice with a checkmark is correct, while the choice with the ChatGPT icon is the one ChatGPT incorrectly selects under zero-shot settings. The *underlined* word is the new term.

- Frequency: Frequency is one of the most important features of terms [1, 28], significantly impact the ability of model [19, 57]. We determine the frequency of a term by obtaining the number of Google Search results prior to the collection period of new terms. To achieve this, we adopt the Custom Search JSON API and use an exact match for the whole new term.

- Deducing Difficulty: To filter out pre-existing or easily deduced terms [21], we use LLMs with specific knowledge cutoffs to deduce the meaning of new terms from their spelling. For terms in 2022, we use `gpt-4-0613` (knowledge cutoff: September 2021), and for terms in 2023, `gpt-4-1106-preview` (knowledge cutoff: April 2023). The deducing difficulty score is calculated as the cosine similarity between the deduced meaning and Gold definitions using Sentence-BERT [58].

- Word/Phrases: Distinct grammatical structures exist between words and phrases [3, 10], making it necessary to discuss them separately. We distinguish whether a term is a word or phrase based on the presence of a space within the term.

We identify new words and phrases as those with the highest deducing difficulty and lowest frequency, and old words with new meanings as those with the highest deducing difficulty and highest frequency. However, when a phrase appears, it rarely acquires new meanings. When we retain terms with the highest 25% frequency and highest 25% deducing difficulty, the proportion of selected words is 12.75%, while for phrases is only 1.66% among all terms. Therefore, we disregard the case of old phrases with new meanings. For instance, new word "Ziziphian" means "Inability to see the goodness in others", new phrase "tall relative" means "An influential patron", and old word "doctor" has a new meaning "To injure (a person or animal) fatally".

## 3.3 Open-Domain Tasks

To evaluate how LLMs handle new terms, we focus on English natural language understanding (NLU) tasks. English is chosen due to its extensive resources and wide usage in the development and evaluation of language models. We design benchmarks for three distinct open-domain tasks in English, each focusing on a fundamental aspect of the abilities of LLMs. For clarity, we have included an example generated by our framework for each task in Figure 3, with more cases given in Appendix A.

**Choice Of Multiple Alternatives (COMA).** To assess the ability of LLMs to *comprehend* new terms from *helpful context*, we focus on natural language inference, which has long been a grand challenge of artificial intelligence [39, 63]. We adopt the methodology outlined by Gordon et al. [27] to construct a causal reasoning task. The primary objective of this task is for the LLMs to identify the most plausible alternative that has a causal relationship with the given premise. In the COMA example, LLMs must first accurately understand the new term "Juggers" in the sentence, which subsequently enables them to provide a correct answer.

**Choice Of Similar Terms (COST).** To assess the ability of LLMs to effectively *utilize* new terms and *distinguish* them from similar ones, we focus on sentence completion, which is a major capability of LLMs [18, 55]. We follow Talmor et al. [60] to create a fill-in-the-blank task. The primary objective of this task is for the LLMs to select the most suitable term to complete the sentence, given a set of choices that encompass both the new terms and those that closely resemble them. In the COST

example, LLMs must demonstrate the ability to coherently incorporate the new term "Ziziphian" into the sentence, thereby completing the sentence accurately.

**Common Sense Judgement (CSJ).** To evaluate the ability of LLMs to *process* and *interpret* new terms in the *absence* of helpful *context*, we focus on commonsense reasoning [15, 68] using a judgment format, implying that the context surrounding the term may not be accurate. We follow Clark et al. [13] and bench authors [8] to develop the judgment task. In this task, we present grammatically correct sentences that incorporate the new term but may not necessarily align with commonsense knowledge. The primary objective is for the LLMs to ascertain the plausibility of the sentence occurring in a realistic scenario. In the CSJ example, the judgment framework makes LLMs fail to deduce the extension meaning of the new term "Tall relative".

## 3.4 Data Generation

We automatically construct questions for these tasks using LLMs, with the input being new terms and their meanings. During the construction process, one challenge is creating high-quality incorrect choices. Directly requesting LLMs to generate incorrect choices may lead to weakly correlated choices that fail to effectively assess the LLMs' understanding. For high-quality choices, we separate the generation of correct and incorrect choices, carefully designing prompts for each. Additionally, to maintain comparability across years, we consistently adopt GPT-4, specifically `gpt-4-0613`, for data generation. Prompts and detailed construction examples are given in Appendix B.

**Question and correct choice generation.** We use LLMs to generate sentences containing new terms and extract questions and correct choices. Detailly, for COMA, sentences must include a fixed phrase, i.e., "As an effect" or "This happened because". The question and its correct choice are generated by dividing sentences at these fixed phrases. For COST, the correct choice is invariably the new term, and the question is the sentence with the correct choice replaced by a blank, denoted as "_". To prevent the dominance of superficial features, i.e., new terms always correct, we also create questions with related terms generated in the "Incorrect Choice Generation" procedure as correct choices with the same approach. For CSJ, the question is identical to the sentence, with the correct choice always being "True". Further, we create questions with "False" as correct choices, by providing the correct question as input and letting LLM modify it to be incorrect.

**Incorrect choice generation.** LLMs are not adept at generating incorrect but semantically related content. To address this issue, we first generate related terms for the new term with slightly different meanings, and then create choices that are correct for these related terms. This obtains incorrect choices closely related to the new term, while avoiding the generation of incorrect content by LLMs.

We first generate **related terms** for each new term by creating a set of terms that partially cover the semantic spectrum of the new term. These terms can be categorized into four groups: 1) Synonyms, 2) Antonyms, 3) Meaning Guessing, which attempt to convey the meaning of the new term using alternative expressions or descriptions, solely based on the spelling of the new term, and 4) Partial Synonyms, which capture only a partial aspect of the meaning of the new term. We collect three terms of each group from LLM responses, then filter out terms that are too similar to each other using Phrase-BERT [66], resulting in a selection of five distinct terms for each new term.

For CSJ, the choices set is simply {True, False}. For the other two multiple-choice tasks, we generate incorrect choices based on the related terms generated here. For COMA, we generate incorrect choices by prompting LLMs to produce choices that are only correct for related terms. We achieve this by replacing the new term in the question with the related term and letting LLMs complete the sentence, which is then considered the incorrect choice. For COST, since it is a fill-in-the-blank task, we directly use the related term set along with the new term as the choices set. For both tasks, the incorrect choices generated by LLMs are not always reasonable, so we generate two extra choices as alternatives, i.e., six choices in total before filtering.

## 3.5 Data Filtering

The incorrect choices LLMs generated are not guaranteed to be incorrect and might also be reasonable. Besides, some questions may also be irrational and do not have a reasonable choice. To tackle this, we prompt LLMs, specifically `gpt-4-0613`, and human annotators with the meaning of the term,

letting them answer questions and filter out inconsistent ones. Finally, we obtain 744 questions for NewTerm 2022, and 715 for NewTerm 2023.

**LLM filtering.** We let LLMs filter the benchmark by prompting them to answer the question we generate, and are allowed to select more than one choice for multiple-choice questions. To minimize bias, we use prompts that differ from those used in the evaluation. Choices and sentences are then filtered under the following conditions: 1) The question is discarded if the prediction does not contain the correct answer, which indicates inconsistency within the question. 2) Then, for multiple-choice questions, we discard incorrect choices that are wrongly identified as correct. 3) If fewer than four choices remain, we discard these questions, as in most cases the answer has low relevance to the term. 4) If more than four choices remain, we eliminate highly similar choices using Sentence-BERT [58] for COMA and Phrase-BERT [66] for COST. Finally, we obtain four-choice questions for COMA and COST, and judgment questions for CSJ. We retain one question per term with higher perplexity calculated by Llama-2-7B [62], which is considered difficult, resulting in 900 questions each year.

**Human filtering.** We adopt human efforts for further verification. One professional and two crowdsource annotators perform human filtering using a clear interactive interface. For multiple-choice questions, we allow users to choose multiple or no choices. For judgment tasks, only True or False is permitted. Finally, in cases of discrepancy among annotators, the final decision is made after a second annotation by the professional annotator. Questions with human answers that do not align with the automatic ones are then filtered out, considered as low-quality questions.

We conducted human annotations on NewTerm 2022 and 2023. Firstly, we calculate the inter-annotator agreement using Fleiss' Kappa [22, 71], which reaches a score of 0.70, indicating substantial agreement with professional annotators. Additionally, in 82.41% of cases, the annotator results match the automatically generated ones, demonstrating the efficiency of our framework for benchmark construction. Finally, out of 900 questions annually, this results in a total of 744 clean questions for NewTerm 2022, and 715 questions for NewTerm 2023, with an overall accuracy rate of 81.06%. Detailed human filtering configurations, as well as consistency analysis of each sub-module and generated data with human annotations, are provided in Appendix C.

**Human filtering can be omitted.** According to the aforementioned human filtering, our automatically generated benchmark has a high quality with over 80% accuracy. Moreover, the evaluation results before and after filtering maintain a high level of consistency, with an average absolute change in accuracy of only 1.59, and the accuracy ranking among LLMs remains completely unchanged. Therefore, human filtering is optional. This significantly reduces the cost of maintaining and updating the benchmark, providing foundation and assurance for our annual real-time updates for the NewTerm benchmark in the future. To alleviate concerns and more accurate result analysis, we report the results under benchmarks after human verification in subsequent experiments.

## 4   Evaluating LLMs on NewTerm

We first analyze the performance of various LLMs in Section 4.2. Subsequently, we analyze the performance variations of LLMs in the dimension of year in Section 4.3 and new term category in Section 4.4. We further analyze why LLMs struggle with new terms in Section 4.5.

### 4.1   Experimental Setup

We evaluate the performance of various LLMs with different knowledge cutoffs, detailed as follows:

**GPT series models.** We evaluate the performance of GPT-4 [51], specifically `gpt-4-0613`, with a knowledge cutoff up to September 2021; `gpt-4-1106-preview`, with a knowledge cutoff up to April 2023; and `gpt-4-0125-preview`, with a knowledge cutoff up to December 2023. We also evaluate ChatGPT [50], specifically `gpt-3.5-turbo-0613` and `gpt-3.5-turbo-0125`, both with a knowledge cutoff up to September 2021. All temperatures are set to 0 while evaluation.

**Claude series models.** We also evaluate `claude-instant-1.2`, a predecessor of Claude Haiku, and `claude-2.1`, a predecessor to Claude 3, both with a knowledge cutoff up to

| LLM | Size | NewTerm 2022 | | | | | NewTerm 2023 | | | | |
|-----|------|------|------|-----|------|------|------|------|-----|------|------|
| | | COMA | COST | CSJ | Avg. | Gold | COMA | COST | CSJ | Avg. | Gold |
| **Llama-2-Chat** | 7B | 28.89 | 28.12 | 60.88 | 39.29 | 58.68 | 32.16 | 33.62 | 83.93 | 49.90 | 64.54 |
| | 13B | 31.24 | 33.19 | 56.11 | 40.18 | 60.92 | 37.72 | 43.08 | 57.50 | 46.10 | 59.19 |
| | 70B | 45.49 | 48.99 | 61.13 | 51.87 | 82.38 | 48.10 | 63.14 | 64.67 | 58.64 | 81.92 |
| **Llama-3-Instruct** | 8B | 52.94 | 46.81 | 63.19 | 54.31 | 88.19 | 54.68 | 67.80 | 70.39 | 64.29 | 91.12 |
| | 70B | 66.01 | 58.70 | 66.15 | 63.62 | 96.07 | 65.35 | 73.59 | 64.94 | 67.96 | 95.83 |
| **Claude-Instant-1.2** | S | 49.28 | 47.54 | 68.60 | 55.14 | 88.33 | 62.28 | 70.48 | 77.03 | 69.93 | 92.18 |
| **Claude-2.1** | M | 38.04 | 54.20 | 71.94 | 54.73 | 82.20 | 41.52 | 64.41 | 82.20 | 62.71 | 83.25 |
| **Claude-3-haiku** | S | 58.04 | 53.62 | 67.18 | 59.61 | 92.60 | 65.20 | 73.31 | 72.78 | 70.43 | 93.52 |
| **Claude-3-sonnet** | M | 56.73 | 56.23 | 64.48 | 59.15 | 93.73 | 65.79 | 70.06 | 67.07 | 67.64 | 94.98 |
| **Claude-3-opus** | L | 64.58 | 67.97 | 65.38 | 65.98 | 93.60 | 72.22 | 79.24 | 60.16 | 70.54 | 93.46 |
| **GPT-3.5-0613** | S | 52.42 | 49.71 | 73.62 | 58.58 | 87.71 | 53.51 | 68.68 | 85.39 | 69.19 | 89.83 |
| **GPT-3.5-0125** | S | 51.37 | 49.86 | 72.07 | 57.77 | 87.63 | 54.82 | 70.06 | 76.36 | 67.08 | 87.90 |
| **GPT-4-0613** | L | 68.37 | 61.16 | 70.14 | 66.56 | 98.91 | 70.18 | 77.01 | 81.01 | 76.07 | 98.72 |
| **GPT-4-1106** | M | 72.03 | 63.48 | 70.79 | 68.76 | 97.56 | 70.32 | 81.21 | 77.16 | 76.23 | 96.34 |
| **GPT-4-0125** | M | 69.80 | 65.94 | 71.94 | 69.23 | 98.11 | 68.86 | 79.94 | 78.49 | 75.76 | 96.59 |
| **Average** | - | 53.68 | 52.37 | 66.91 | 57.65 | 87.11 | 57.51 | 67.71 | 73.27 | 66.16 | 87.96 |

Table 1: Main results for different LLMs under NewTerm 2022 and 2023. The definitions of "COMA", "COST" and "CSJ" can be found in Section 3.2, while "Base" and "Gold" in Section 4.1. "S", "M", and "L" represent small, medium, and large, respectively, inferred based on the API pricing.

early 2023 [2]. Further, we evaluate all sizes of Claude 3, i.e., `claude-3-haiku-20240307`, `claude-3-sonnet-20240229`, and `claude-3-opus-20240229`, with model size from small to large, all have a knowledge cutoff up to August 2023. All temperatures are set to 0 while evaluation.

**Llama series models.** We evaluate Llama-2-chat 7B, 13B, and 70B [62], with a knowledge cutoff up to September 2022. We also evaluate Llama-3-Instruct 8B, with a knowledge cutoff up to March 2023, and Llama-3-Instruct 70B, with a knowledge cutoff up to December 2023. All tests are done under greedy decoding.

**Prompts.** According to preliminary experiments, the few-shot settings do not show obvious improvements. Thus, we test LLMs in zero-shot settings without providing any additional information (**Base**) to evaluate the ability to understand new terms, and zero-shot settings with the meaning of the term prompted (**Gold**) to assess the inherent capabilities of LLMs. Subsequently, we consider the performance gap between Base and Gold settings as the performance decline caused by new terms. We run each prompt once, and any failure to answer is deemed an error, which occurs infrequently during evaluation (<2% on average). In most of these cases, they refuse to answer because they do not know the new terms.

## 4.2 Main Results

Results are in Table 1. Using `gpt-4-0613` for filtering may introduce bias, resulting in an overestimation of the performance of GPT series models, especially for `gpt-4-0613` itself. However, the relative value between Base and Gold remains meaningful. Additionally, despite their higher performance, we can still draw the following conclusions. To support these results, we conduct experiments on more open-source LLMs, with results showing similar trends, detailed in Appendix D.

**New terms are challenging for LLMs.** Results under Gold settings can be seen as the score for LLMs when they understand every term in the question. Compared to Gold setting, Base setting results in consistently and significantly worse performance (-25.63 on average), thus proving the significant performance decrease caused by new terms not known by LLMs. Results under Gold settings can also be seen as the estimation of the upper bound for each LLM using prompt-based improvement methods, thus proving the great potential for further improvement.

**Larger LLMs lead to higher performance but less impact on performance decrease with new terms.** We compare LLMs of varying sizes, specifically examining the largest and smallest versions of each series of models released at the same time. We observe that apart from `claude-instant-1.2` and `claude-2.1` which are not strictly the same version models, other LLMs exhibit superior

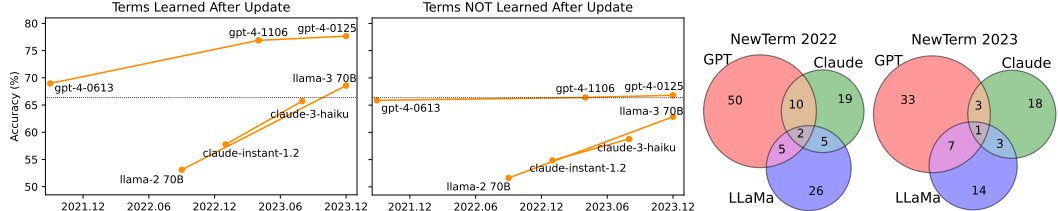

Figure 4: (Left) Performance of LLMs on different terms under Base setting in NewTerm 2022. The dashed line in the middle figure represents the average accuracy of GPT-4. (Right) The overlap of learned terms selected by each series of models in NewTerm 2022 and 2023.

performance under Gold settings, achieved an average improvement of +9.39, demonstrating a stronger ability under current tasks. However, the average performance decrease caused by new terms (Base - Gold) in larger LLMs even shows a slight increase (-25.13 vs -26.94 for smaller and larger LLMs, respectively). This suggests that powerful LLMs still struggle to address new terms.

**Performance differences under new terms across years.**    With a unified construction setting, there is no significant performance change under Gold setting in NewTerm 2023 compared to 2022 (+0.85 on average). However, a noticeable increase was observed under Base setting (+8.51 on average). This suggests that new terms from recent years are not necessarily more challenging. Additionally, as the knowledge cutoff is updated, the performance improvement in 2022 is significantly more pronounced. For `claude-instant-1.2` and `gpt-4-0613`, minor changes occur under Gold setting after updating to `claude-3-haiku` and `gpt-4-0125` (+0.67). However, under Base setting, the changes are +3.57 vs +0.10 for NewTerm 2022 and 2023, respectively. This indicates that updates have a more noticeable impact on improving new terms within the knowledge cutoff.

## 4.3   Results for Terms and LLMs of Different Years

Despite observing upward trends with updates on NewTerm 2022, the trends are not significant. Therefore, we assert that LLMs do not learn all new terms after updates. To demonstrate this, we extract and analyze new terms that LLMs have indeed learned after the update.

**Selection of learned new terms.**    We select learned terms by comparing the deduce difficulty across LLMs with different knowledge cutoffs. LLMs are first asked to deduce the meaning of new terms from their spelling. The difficulty score is calculated as the cosine similarity between the deduced meaning and Gold definitions, using Sentence-BERT [58]. We then filter out the hardest 1/3 terms with the lowest similarity in the newer LLMs, considering them unlearned. Learned terms are selected if they show a similarity increase of over 15% compared to the older LLMs.

Although this method is simplistic and cannot guarantee to select all learned terms, it still yields satisfactory results. To demonstrate this, we conduct experiments under NewTerm 2022, which is within the knowledge cutoff of the latest LLMs. For each series of LLMs, we first use the oldest and newest LLMs to classify new terms as learned and unlearned. This classification is then used to evaluate LLMs of this series, with results under Base setting on the left of Figure 4. Compared to unlearned terms, learned terms exhibit substantial improvements after knowledge cutoff updates (+10.70 vs +5.37 for learned and unlearned terms, respectively). We present cases for the learned terms and their corresponding downstream task performance in Appendix E.

**Parts of new terms within the knowledge cutoff of LLMs are learned.**    Using the above methods, we found that there are 50% more new terms learned in 2022 compared to 2023 (38, 36, 67 vs 25, 25, 44 for Llama, Claude, and GPT respectively). Considering that the newer LLMs' knowledge cutoff is in mid-to-late 2023, results demonstrate that LLMs can update new terms within their knowledge cutoff but are hard to generalize to terms from more recent periods.

**Limited overlap in learning new terms across different models.**    We evaluate the degree of overlap for learned terms selected by different LLM series under NewTerm 2022 and 2023. As shown in the right of Figure 4, there is limited overlap between learned terms selected by different series. Additionally, learned terms selected by other series of LLMs exhibit limited performance

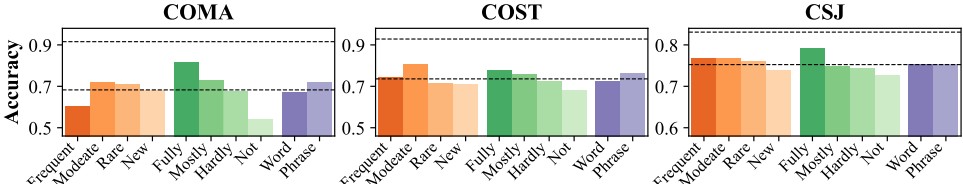

Figure 5: The performance of ChatGPT for different types of new terms. Orange columns represent frequency, green represents deducing difficulty, and purple represents Word/Phrase. The lower dashed line represents the average score for Base setting, while the higher one represents Gold setting.

improvement compared to unlearned ones. These findings indicate that the overlap of new terms learned by different series of LLMs is limited. It is worth noting that the knowledge cutoff spans vary among series of LLMs, which also contributes to the limited overlap among them.

## 4.4 Results for Terms of Different Category

To test which new terms and questions are more challenging, we automatically constructed a new ablation benchmark with `gpt-4-0613` based on terms in 2022 with no human effort. To comprehensively evaluate different types of new terms, we remove the "New Term Selection" procedure and randomly select 1.2k new terms. We then generate more questions per term without discarding by perplexity. Finally, we obtain 6.6k questions for COMA, 6.2k for COST, and 7.1k for CSJ.

Then, we categorize terms across three dimensions defined in Section 3.2: 1) frequency from high to low: Frequent, Moderate, Rare, and New and 2) deducing difficulty from low to high: Fully, Mostly, Hardly, and Not deduced. Except for "New" terms, which are defined as terms with fewer than ten results in Google Search, other categories are evenly split. We also test results of 3) Word and Phrase.

**Frequency and deducing difficulty are strongly correlated with performance.** We report the average score of terms in each category in Figure 5. We observe a positive correlation between the frequency of terms and their accuracy, and in COMA and COST, "Frequent" terms also tend to perform poorly. These suggest that LLMs sometimes struggle to comprehend new terms and new meanings for frequent terms. Besides, deducing difficulty and accuracy are strictly positively correlated, suggesting that terms harder to deduce are also more challenging for LLMs to comprehend. Finally, phrases consistently yield higher accuracy, implying that phrases tend to be more easily understood than words. The terms with the highest and lowest frequencies, as well as the highest deducing difficulty, correspond precisely to the three types of new terms we are investigating: new words, new phrases, and old words with new meanings.

## 4.5 Why LLMs Struggle in New Terms?

We randomly selected 135 cases where ChatGPT, specifically `gpt-3.5-turbo-0613`, failed under zero-shot settings in NewTerm 2022, averagely separated for each type of term and each task. We summarize the following three main types of errors, with cases shown in Figure 3.

**Ignoring new terms.** LLMs sometimes ignore the new term and choose the answer based on other parts of the question. In the COMA case, ChatGPT chooses the answer that focused only on the information about online shopping, ignoring the information that the new term may carry.

**Not preferring to use new terms.** LLMs do not tend to use new terms to complete the sentence, even when no suitable choice is available. In the COST case, even when other choices are all unsuitable, ChatGPT chooses the word "blind", which is grammatically incorrect but partially reasonable.

**Incorrectly understanding new terms.** LLMs sometimes incorrectly understand the new term and make wrong inferences. In the CSJ case, ChatGPT mistakenly understands the phrase "tall relative" in the literal sense as a relative with a high height, leading to the misjudgment.

**Quantitative analysis.** To show which case is more common under different conditions, we counted the error number of different types, as demonstrated in Figure 6. We can see that:

- For *COMA*, ignoring is more common. This is because our pipeline ensures that, in most cases, when ignoring the new term, a reasonable choice is also available.

- For *COST*, in most failed cases, the answer is the new term, but LLMs choose old words that are not reasonable. Only one case is observed where the answer is not a new term.

- For *CSJ*, misunderstandings are more common since LLMs need to judge the whole sentence, and they try to understand the new term more. Sometimes, they may also regard the new term as a spelling error and consider it as incorrect instead of ignoring it.

- For new phrases (*NewP*), LLMs are more likely to misunderstand due to more semantic information in the spelling, while for new words (*NeW*), ignoring is more common. old words with new meanings (*OldW*) is in between.

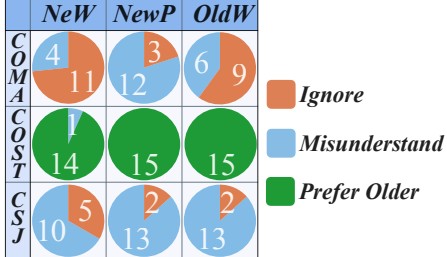

Figure 6: A quantitative analysis of error reasons for ChatGPT under the zero-shot setting. "Ignore" means ignoring new terms, "Misunderstand" means incorrectly understanding new terms, and "Prefer Older" means not preferring to use new terms.

## 5 Conclusions and Limitations

We proposed a highly automated method for benchmarking real-time new terms, i.e., NewTerm. Experiments on various LLMs highlight the challenges they face in understanding new terms. Three types of terms pose more significant challenges: new words, new phrases, and old words with new meanings. Additionally, while updates to the knowledge cutoff of LLMs can cover some new terms, they are unable to generalize to more distant ones. We have released NewTerm 2022 and 2023, and will continue to update them annually to track the performance of LLMs.

**Limitations.** This paper has several limitations.

- Although our framework is not dependent on any specific LLM, it demands high performance from them. To partially alleviate concerns about reproducibility, we conduct further experiments by employing two different LLMs, i.e., `gpt-4-0613` and `claude-2.1`, to generate benchmarks. Human annotations and experimental results confirm the high validity of both of the benchmarks, with detailed analysis in Appendix F.

- It is hard to control variables between benchmarks of different years, as the collected new terms often have varying numbers and distributions, making comparisons of term difficulty across years difficult. To mitigate this issue, we use the same settings when generating benchmarks for evaluation.

- The highly automated and cost-effective construction pipeline offers substantial value in evaluating LLMs' understanding of a broader range of terms. However, our method has not been validated across a wider variety of new terms, with coverage currently limited to 300 new terms per year, which may introduce potential bias. Additionally, our approach has only been validated using English online dictionaries. On one hand, our method has the potential to be extended to new terms from broader sources, such as online forums and specialized domains. On the other hand, for multilingual new terms, our approach could be effectively adapted with minimal prompt modifications. However, due to budget constraints, we were unable to conduct validation across more diverse and extensive term sources.

## Acknowledgments

This work was supported in part by the National Natural Science Foundation of China (Grant No. 62206076), Guangdong Basic and Applied Basic Research Foundation (Grant No. 2024A1515011491), Shenzhen Science and Technology Program (Grant Nos. ZDSYS20230626091203008, KJZD20231023094700001, RCBS20221008093121053), and Shenzhen College Stability Support Plan (Grant Nos. GXWD20220811173340003, GXWD20220817123150002). Xuebo Liu was sponsored by CCF-Tencent Rhino-Bird Open Research Fund. We would like to thank the anonymous reviewers and meta-reviewers for their insightful suggestions.

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

# A Task-Level Case Study

This section illustrates how a model's performance may vary across different tasks associated with the same new term. We analyzed the performance of Llama-3-Instruct-70B on the new term "wokely," defined as an adjective meaning "Of little worth; poor, mean, paltry." The model's performance varied across three tasks under the zero-shot Base setting:

| Task | Question | Response |
|------|----------|----------|
| COMA | The book's cover was described as wokely by several reviewers. I am hesitating among these options. Help me choose the more likely effect:
A. it struggled to attract attention on the bookstore displays despite a compelling narrative inside.
B. many readers were enticed to buy it, strengthening its presence on the bestseller list.
C. readers were intrigued and the book's sales experienced an unexpected surge worldwide.
D. the publisher decided to release a limited edition with a special hardback velvet cover. | A (✓) |
| COST | The goods at the flea market appeared distinctly _, making it hard to find a satisfying purchase. In the previous sentence, does _ refer to A. Spokely, B. Cokely, C. Wokely, or D. Worthy? | D (X) |
| CSJ | His contributions to the project were considered wokely, barely making any impact. Is this example in line with commonsense and grammatically correct? | Incorrect (X) |

Table 2: Performance of Llama-3-Instruct-70B on Different Tasks Involving the New Term "wokely"

As observed, the model only answered correctly in the COMA task but failed in the other two tasks. In the COMA task, the model successfully inferred that "wokely" carries a negative connotation, allowing it to correctly choose choice A. This demonstrates its ability to *comprehend* the new term within a *helpful* context. However, in the COST task, where the model needed to *utilize* the new term and *distinguish* it from similar choices, it struggled. Although the phrase "hard to find a satisfying purchase" suggested the need for a negative term, the model incorrectly chose "Worthy," which is grammatically correct but semantically incorrect. In the CSJ task, the model was required to *process* and *interpret* the new term in the *absence* of helpful *context*. The context matched the definition of "wokely" perfectly, yet the model erroneously judged the response as incorrect because it was a judgment-based evaluation.

These results provide a comprehensive evaluation of the model's understanding of the term "wokely." They reveal that while the model can recognize that it is a negative term when the context is clear, it struggles to grasp the detailed meaning of the term and how to accurately use it in different contexts.

# B    Benchmark Generation Cases and Prompts

**Benchmark generation cases.**    For clarity, we provide cases to illustrate how to extract questions and correct choices from sentences, as shown in Figure 7. In these examples, the two cases from COMA correspond to the inclusion of fixed phrases "As an effect" and "This happened because", respectively. The two cases from COST represent the new term and its related term as the correct answers, respectively. The two cases from CSJ correspond to questions with answers being True and those with modified answers being False, respectively.

Furthermore, we provide an example of the COMA task construction process, as shown in Figure 8. Ultimately, we filter out choices A and E, resulting in the final clean question being the current question, along with a multiple choice question that contains only choices B, C, D and F.

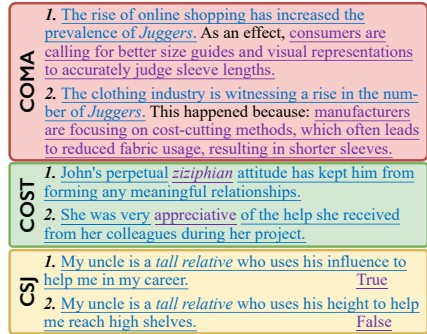

Figure 7: Examples of question and correct choice generation. We first generate sententence, then separate it to obtain the question and correct choice.

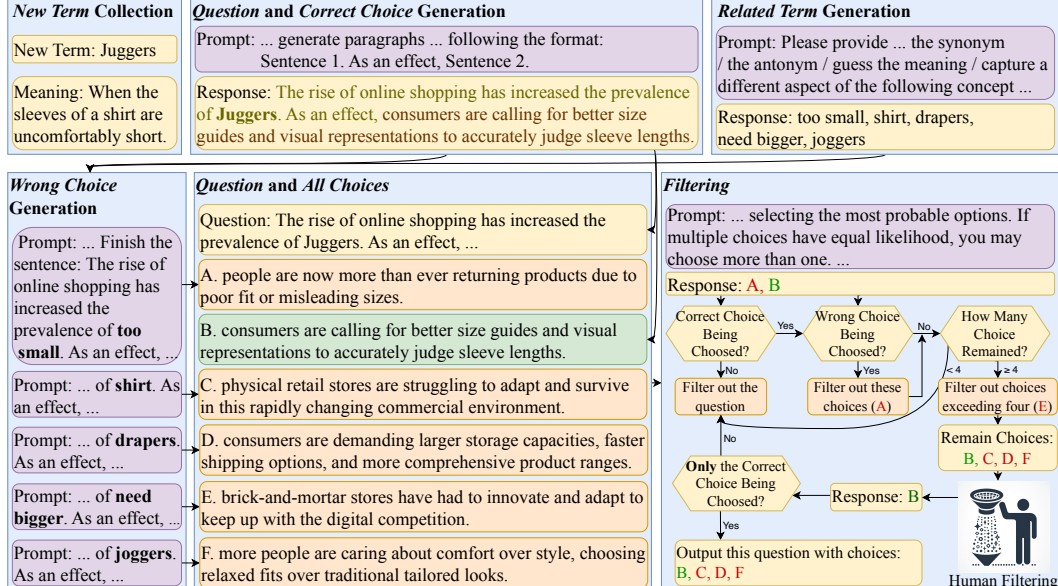

Figure 8: An example of the COMA task construction process. The input is the collected new term and its meaning, and the output is the question with choices B, C, D, and F, where B is the correct choice.

**Benchmark generation prompts.**    Further, we introduce the prompt we used in benchmark construction and LLM evaluation. We use "[·]" to express variables depending on the input. The notation "[W]" represents the new term and "[M]" represents the meaning of the term. We use "[Ti]" to represent the $i$-th related term of the new term, "[Ci]" to represent the $i$-th choice we generated, and "[N]" to represent the number of questions we need to generate per term. We use an underline to show we use only one of the choices separated with "/". Additionally, LLMs do not always generate valid outputs. For cases where we do not get enough outputs, we generate multiple times until we obtain enough distinct outputs.

- For procedure "New Term Collection", we use LLMs to get the deduce difficulty of each term. Prompts are detailed in Table 3.
- For procedure "Question and Correct Choice Generation", we use LLMs to get different types of sentences for each of the three tasks. Prompts are detailed in Table 4, Table 5, and Table 6.

- For procedure "Related Term Generation", we use LLMs to get four different types of related terms for each new term. Prompts are detailed in Table 7, Table 8, and Table 9.

- For procedure "Incorrect Choice Generation", we only need LLMs to generate incorrect choices for task COMA. Prompts are detailed in Table 10.

- For procedure "LLM Filtering", we use LLMs to filter all the benchmarks of the three tasks. Prompts are detailed in Table 11, Table 12, and Table 13.

- For evaluation, we follow the prompt of similar datasets in PromptSource [5] to design five prompts manually and select three that have the highest performance for ChatGPT under Gold settings from them. Prompts are detailed in Table 14, Table 15, and Table 16.

| Deduce Difficulty | |
| --- | --- |
| System Prompt | Please deduce the meaning of the following word based on its spelling, using just one sentence. |
| User Prompt | What is the meaning of "[W]"? Meaning: |

Table 3: Prompt for the Deduce Difficulty.

| Sentence Generation | |
| --- | --- |
| System Prompt | Please generate [N] different sentences about the new term, each in a separate line, without using the words used above. Make sure that all the sentences you generate have a different subject. Please print the sentence without explanation. |
| User prompt for COMA | I create a new term "[W]", which means "[M]". Please generate [N] different paragraphs about "[W]", following the format: "Sentence 1. As an effect, / This happened because: Sentence 2." Sentence 1 should contain "[W]" once. Ensure that it is objective and impartial, focusing on actual actions or events, without any emotional or subjective assumptions. Sentence 2, illustrating the effect / cause of Sentence 1, should be specific to "[W]" in Sentence 1 and not applicable if "[T1]", "[T2]", "[T3]", or "[T4]" is used instead. |
| User prompt for COST & CSJ | I have created a new term, "[W]", which means "[M]". Please generate [N] different sentences about "[W]", each in a separate line, which should be specific to the meaning of "[W]". The sentence should be grammatically correct but not applicable if "[T1]", "[T2]", "[T3]", or "[T4]" is used instead. |

Table 4: Prompt for the Sentence Generation. We generate $N$ sentences simultaneously for each new term to reduce costs.

| Sentence Generation for the Second Half of COST | |
| --- | --- |
| System Prompt | Please generate a sentence about the term "[Ti]", without using the words used above. Make sure that "[Ti]" is exactly in the sentence but not its other forms. Please print the sentence without explanation. |
| User prompt | Please generate a sentence about "[Ti]", which should be specific to the meaning of "[Ti]". The sentence should be grammatically correct but not applicable if "[T1]", "[T2]", "[T3]", or "[T4]" is used instead. Sentences: |

Table 5: Prompt for the Sentence Generation for the Second Half of COST. For each generated sentence, we assign different related terms as the answer.

| **Sentence Generation for the Second Half of CSJ** | |
| --- | --- |
| System Prompt | Please generate [N] different sentences about the new term, each in a separate line, without using the words used above. Make sure that all the sentences you generate have a different subject. Please print the sentence without explanation. |
| User prompt | For each sentence generated above, please modify it to use "[W]" illogically, based on the given meaning, while keeping the grammar, fluency, and original subject intact. For each example, print "Wrong Sentence:" and "Corresponding Wrong meaning:" on separate lines, explaining the deviation from the intended meaning. Ensure that each wrong meaning is significantly different from those previously generated. |

Table 6: Prompt for the Sentence Generation for the Second Half of CSJ. The user prompt and response of correct sentence generation for CSJ are also used as context input.

| **Partial Synonym Term Generation** | |
| --- | --- |
| System Prompt | Please provide three words and three two-word phrases, and display each of them on a separate line. The first three lines are words, each on a separate line, and the last three lines are phrases, each on a separate line. Make sure that there are six lines in total, with each word/phrase at a single line. Do not refrain from answering. |
| User prompt | Please provide three words and three phrases, "[M]". Ensure that these are commonly used and easily understood by a 3-year-old child. |

Table 7: Prompt for the Partial Synonym Term Generation.

| **Synonym & Antonym Term Generation** | |
| --- | --- |
| System Prompt | Please answer the following question by printing three terms without explanation, each at a separate line. If you cannot construct terms that fully meet the requirements, provide terms that partially fulfill the requirements. Do not refrain from answering. |
| User prompt | What is the synonym / antonym for the new term, "[W]", that refers to [M]? The synonym / antonym should be a commonly used English term and belong to the same part of speech. Do not use abbreviations and commas, periods in the term, and shorter than five words. Please generate three different alternatives. Synonym / Antonym: |

Table 8: Prompt for the Synonym and Antonym Term Generation.

| **Meaning Guessing Term Generation** | |
| --- | --- |
| System Prompt | Please answer the following question by printing three terms without explanation, each at a separate line. If you cannot construct terms that fully meet the requirements, provide terms that partially fulfill the requirements. Do not refrain from answering. |
| User prompt | Please guess the meaning of the term "[W]" and create three alternative terms based on their spelling. Alternative term: |

Table 9: Prompt for the Meaning Guessing Term Generation.

**COMA Incorrect Choice Generation**

| | |
|---|---|
| System Prompt | Please generate a sentence with ... words to finish the following paragraph. Please print the sentence without explanation. |
| User prompt | [Replace the new term [W] in [Question] with its related term [Ti]]. As an effect, / This happened because: |

Table 10: Prompt for the COMA Incorrect Choice Generation. For each generated question, we create completions that are correct for each related term as incorrect choices. To make it more challenging to distinguish, we prompt that the lengths of the incorrect choices generated by LLM are as close as possible to the correct ones.

**LLM Filtering for COMA**

| | |
|---|---|
| System Prompt | Please answer the following choice question by selecting the most probable choices. If multiple choices have equal likelihood, you may choose more than one. List the selected choices (A, B, C, D, E, or F) separated by commas. |
| User prompt | Given that the term "[W]" means "[M]", please solve the following multiple-choice exercise: Exercise: choose the most plausible alternative. [Question] so / because... A. [C1] B. [C2] C. [C3] D. [C4] E. [C5] F. [C6] Answer: |

Table 11: Prompt for the LLM Filtering for COMA.

**LLM Filtering for COST**

| | |
|---|---|
| System Prompt | Please answer the following choice question by selecting the most probable choices. If multiple choices have equal likelihood, you may choose more than one. List the selected choices (A, B, C, D, E, or F) separated by commas. |
| User prompt | Given that the term "[W]" means "[M]", please solve the following multiple-choice exercise: [Question] Replace the __ in the above sentence with the correct choice: A. [C1] B. [C2] C. [C3] D. [C4] E. [C5] F. [C6] Answer: |

Table 12: Prompt for the LLM Filtering for COST.

**LLM Filtering for CSJ**

| | |
|---|---|
| System Prompt | Please answer the following question with an integer, without any further explanation. |
| User prompt | Given that "[W]" means "[M]". On a scale of 0 to 10, with 0 being extremely unlikely and 10 being highly likely, how probable is it that the following sentence is coherent and aligns with general understanding? [Question] Answer: |

Table 13: Prompt for the LLM Filtering for CSJ.

**COMA Evaluation**

| | |
|---|---|
| System Prompt for Base Setting | Please answer the following question by printing exactly one choice from "A", "B", "C", "D", without explanation. |
| System Prompt for Gold Setting | Given that "[W]" means "[M]". Please answer the following question by printing exactly one choice from "A", "B", "C", "D", without explanation. |
| User prompt 1 | Exercise: choose the most plausible alternative. [Question] because / so... A. [C1] B. [C2] C. [C3] D. [C4] Answer: |
| User prompt 2 | [Question] In the previous sentence, does __ refer to A. [C1], B. [C2], C. [C3], or D. [C4]? Answer: |
| User prompt 3 | Fill in the __ in the below sentence: [Question] Choices: A. [C1] B. [C2] C. [C3] D. [C4] Answer: |

Table 14: Prompt for the COMA Evaluation.

**COST Evaluation**

| | |
|---|---|
| System Prompt for Base Setting | Please answer the following question by printing exactly one choice from "A", "B", "C", "D", without explanation. |
| System Prompt for Gold Setting | Given that "[W]" means "[M]". Please answer the following question by printing exactly one choice from "A", "B", "C", "D", without explanation. |
| User prompt 1 | [Question] Replace the __ in the above sentence with the correct choice: A. [C1] B. [C2] C. [C3] D. [C4] Answer: |
| User prompt 2 | [Question] Is this example in line with commonsense and grammatically correct? Answer: |
| User prompt 3 | Given that "[W]" means "[M]". On a scale of 0 to 10, with 0 being extremely unlikely and 10 being highly likely, how probable is it that the following sentence is coherent and aligns with general understanding? [Question] Answer: |

Table 15: Prompt for the COST Evaluation.

**CSJ Evaluation**

| | |
|---|---|
| System Prompt under Base Setting | Please answer the following question by printing "YES / Acceptable" or "NO / Unacceptable", without explanation. |
| System Prompt under Gold Setting | Given that "[W]" means "[M]". Please answer the following question by printing "YES / Acceptable" or "NO / Unacceptable", without explanation. |
| User prompt 1 | Does the following sentence coherent and aligned with general understanding? Please answer "YES" or "NO". [Question] Answer: |
| User prompt 2 | [Question] Is this example in line with commonsense and grammatically correct? Answer: |
| User prompt 3 | The following sentence is either "Acceptable", meaning it fits the commonsense, or "Unacceptable". Which is it? [Question] Answer: |

Table 16: Prompts for the CSJ Evaluation.

# C Human Filtering

## C.1 Human Filtering Settings

**Interactive interface.**   Our human-interactive interface, built using the SurveyJS library in Vue3 frontend and Flask backend, is designed to provide a user-friendly workflow and efficient annotator experience for our new term benchmark. The platform supports translation, flexible question numbers, and loading history for all three question types. By translating questions into the native language of annotators and providing a clear interface, users can answer questions in about 30 seconds, completing annotations for 900 questions in 10 hours.

Upon accessing the platform, users receive a welcoming message and need to fill in a unique username, ensuring each user can only fill out one questionnaire, as shown in Figure 9. The platform also allows users to decide the total number of questions they wish to answer. The "Loading History" feature enables users to load and modify their previous history. Choosing "Yes" includes all previously answered questions in their total count and allows users to check and change previous answers, while selecting "No" provides new questions.

On the answering page, our interface comprises three question types, as shown in Figure 10. We separate different types of questions into distinct pages, with each page containing 10 questions. Answers are saved after annotators finish any page, making it easy for them to skip and return to continue at any time. Finally, to support situations with no choices and to provide feedback and records for special cases, we have set up two additional choices, namely "None" and "Other".

**Annotators.**   For human filtering, we recruited two crowdsource annotators and one professional annotator.  For the crowdsource annotators, we enlisted the services of two English-proficient annotators from China via a crowdsourcing platform. After evaluation, we determined the annotation cost to be RMB 1.5 per question per person. For the professional annotator, we engaged a university professional annotator, who is a current master's student specializing in natural language processing, to perform the annotation.

To minimize inconsistencies, we provide users with detailed guidance, including annotation instructions, examples, and requirements. Specifically, for multiple-choice questions, annotators are asked to select the choice that best aligns with the question's intent.  If multiple choices have similar probabilities and are all reasonable, they should select multiple choices. If none of the choices are reasonable, they should choose "None". Based on our evaluation and filtering experience with LLMs on NewTerm, we observed that these annotation criteria closely resemble the standards used for most LLMs. Since our benchmark aims to evaluate the performance of LLMs, we chose criteria for human annotation that align as closely as possible with LLMs.

Additionally, to increase efficiency and reduce annotation costs, we provide translations of the questions. To minimize bias introduced by translation, we require annotators to be proficient in English during the recruitment process.  We also emphasize in our instructions that translations may be inaccurate due to the presence of new terms and ask annotators to use translations only for supplementary understanding while basing decisions solely on the English question. Our final decision is made by the professional annotator with strong English reading and writing skills, who can better adhere to our requirements. This approach helps minimize potential risks of errors and ambiguities while achieving lower annotation costs and higher annotation efficiency.

## C.2 Analysis of Human Filtering

**Filtering reason analysis.**   We analyze the reasons for answers that do not align with the three human annotations under NewTerm 2022 and 2023, i.e., humans choosing more than one choice (Multi.), no choices (Zero), or choosing choices differing from auto-generated ones (Wrong). Results are in Table 17. In our construction pipeline, "Multi." is caused by LLM filtering, which failed to choose all the incorrect

|  | Multi. | Zero | Wrong | Acc. (%) |
|---|---|---|---|---|
| **COMA** | 102 | 112 | 202 | 76.89 |
| **COST** | 129 | 142 | 77 | 80.67 |
| **CSJ** | - | - | 281 | 84.39 |

Table 17:  The number of cases where the automatically generated answer does not align with human annotation. "Acc." denotes the percentage of non-alignments, with "Multi.", "Zero", and "Wrong" denotes the number of errors defined in Appendix C.2.

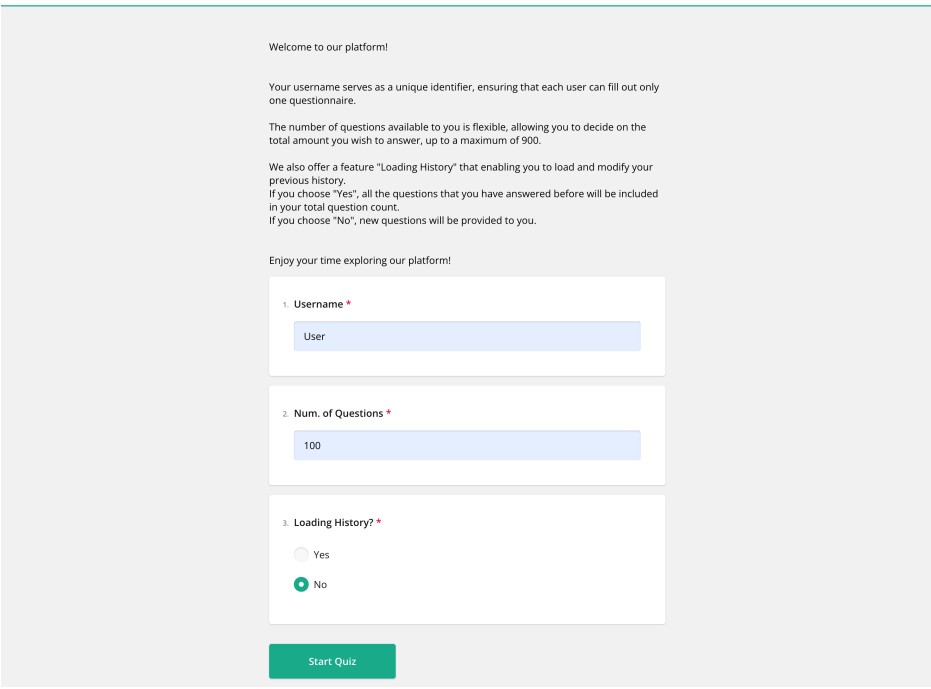

Figure 9: Welcome page of the human-interactive interface, displaying a welcoming message and choices for loading history and selecting the number of questions.

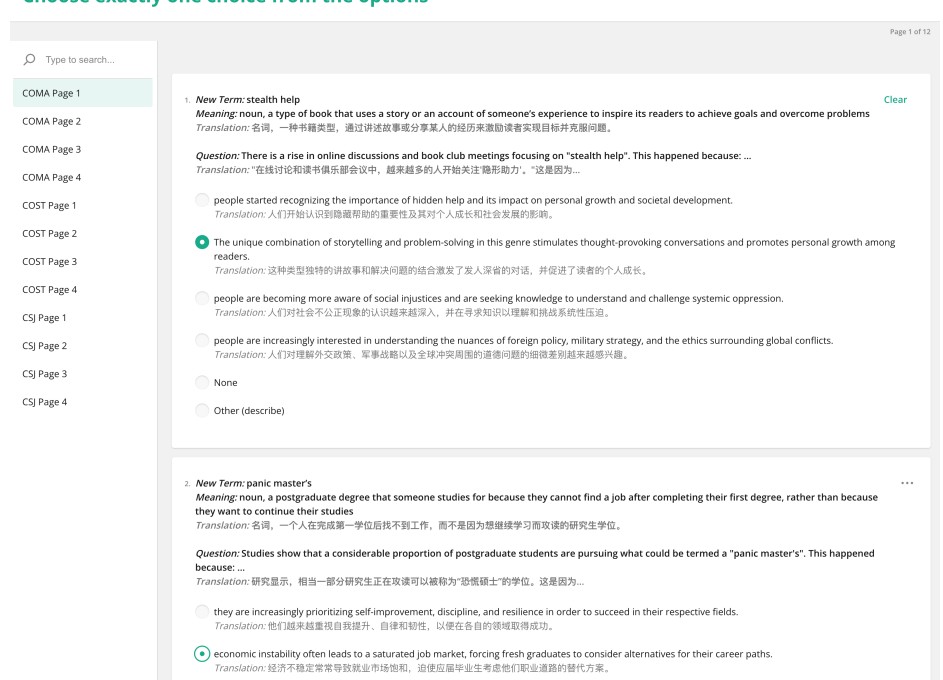

Figure 10: Answering page of the human-interactive interface, showcasing the three tasks in NewTerm benchmark: COMA, COST, and CSJ.

choices that are reasonable, covering 22.11% of the incorrect cases. "Zero" is caused by question generation, where LLMs do not understand the new term correctly and generate meaningless questions or incorrect answers. All errors in CSJ are also caused by this reason. It covers 51.19% of the cases. "Wrong" means that both are partly incorrect; the correct answer is not entirely correct, and LLMs fail to choose all choices that are more plausible than the correct answer. This covers 26.70% of the cases. Stronger LLMs may further alleviate this problem and make the pipeline more reliable.

**Subprocess analysis.**    As a cascaded generation benchmark, error propagation can often occur between subprocesses, making it necessary to analyze the error rates for each subprocess. In our framework, there are two types of error propagations in these steps:

- First, the results of "Related Term Generation" are used for "Incorrect Choice Generation" of COMA and COST questions with answers being old terms. However, these COST questions aim to generate fill-in-the-blank questions related to old terms. As long as a valid term is generated, valid questions can still be generated.

- Second, the results of "Question and Correct Choice Generation" are used for "Incorrect Choice Generation" of COMA and CSJ questions with the answer "False". However, these CSJ questions aim to generate incorrect sentences in the judgment task. Even if the first part of the sentence is not correct, valid incorrect sentences can still be generated.

Therefore, the error propagation problem mainly occurs in the generation of the COMA dataset. To further quantitatively assess the impact of error propagation problems, we randomly select 50 cases of the COMA task in NewTerm 2022 for human annotation. The "Related Term Generation" procedure has a 7.60% error probability, where the generated term is less related to the new term. The "Question and Correct Choice Generation" procedure has a 12.00% error probability, where the generated sentence is incorrect for the new term.

The "Incorrect Choice Generation" procedure is based on the output of the above procedures. Additionally, incorrect questions should be discarded regardless of the choices, so we ignore cases with incorrect questions in the subsequent annotation process. Two types of errors occur in incorrect choice generation: 1) First, the generated incorrect choices are reasonable under the current question, covering 26.89% of choices. 2) Second, due to error propagation from the related term, the choice may be irrelevant to the original question. However, we did not observe this phenomenon in the 264 annotated choices with valid questions. This is because the question occupies the main part of the prompt, and a single irrelevant term is not enough to interfere with LLMs to generate irrelevant choices.

| LLM | Size | NewTerm 2022 w/ human filtering | | | | | NewTerm 2022 w/o human filtering | | | | |
|---|---|---|---|---|---|---|---|---|---|---|---|
| | | COMA | COST | CSJ | Avg. | Gold | COMA | COST | CSJ | Avg. | Gold |
| | 7B | 28.89 | 28.12 | 60.88 | 39.29 | 58.68 | 31.56 | 28.89 | 58.67 | 39.70 | 56.33 |
| **Llama-2-Chat** | 13B | 31.24 | 33.19 | 56.11 | 40.18 | 60.92 | 30.78 | 33.56 | 57.11 | 40.48 | 58.67 |
| | 70B | 45.49 | 48.99 | 61.13 | 51.87 | 82.38 | 45.11 | 51.33 | 61.67 | 52.70 | 78.48 |
| **Llama-3-Instruct** | 8B | 52.94 | 46.81 | 63.19 | 54.31 | 88.19 | 51.67 | 51.67 | 63.78 | 55.70 | 85.41 |
| | 70B | 66.01 | 58.70 | 66.15 | 63.62 | 96.07 | 66.78 | 62.33 | 67.00 | 65.37 | 94.85 |
| **Claude-Instant-1.2** | S | 49.28 | 47.54 | 68.60 | 55.14 | 88.33 | 49.56 | 52.00 | 68.22 | 56.59 | 86.22 |
| **Claude-2.1** | M | 38.04 | 54.20 | 71.94 | 54.73 | 82.20 | 37.89 | 56.44 | 70.67 | 55.00 | 79.22 |
| **Claude-3-haiku** | S | 58.04 | 53.62 | 67.18 | 59.61 | 92.60 | 58.89 | 57.67 | 68.00 | 61.52 | 90.56 |
| **Claude-3-sonnet** | M | 56.73 | 56.23 | 64.48 | 59.15 | 93.73 | 56.22 | 58.33 | 65.56 | 60.04 | 92.19 |
| **Claude-3-opus** | L | 64.58 | 67.97 | 65.38 | 65.98 | 93.60 | 64.78 | 70.00 | 67.00 | 67.26 | 92.85 |
| **GPT-3.5-0613** | S | 52.42 | 49.71 | 73.62 | 58.58 | 87.71 | 52.89 | 53.56 | 72.67 | 59.70 | 85.30 |
| **GPT-3.5-0125** | S | 51.37 | 49.86 | 72.07 | 57.77 | 87.63 | 52.56 | 54.44 | 71.33 | 59.44 | 84.78 |
| **GPT-4-0613** | L | 68.37 | 61.16 | 70.14 | 66.56 | 98.91 | 70.78 | 65.22 | 70.33 | 68.78 | 98.59 |
| **GPT-4-1106** | M | 72.03 | 63.48 | 70.79 | 68.76 | 97.56 | 71.78 | 67.22 | 71.11 | 70.04 | 97.11 |
| **GPT-4-0125** | M | 69.80 | 65.94 | 71.94 | 69.23 | 98.11 | 70.33 | 68.78 | 72.56 | 70.56 | 97.70 |
| **Average** | - | 53.68 | 52.37 | 66.91 | 57.65 | 87.11 | 54.11 | 55.43 | 67.05 | 58.86 | 85.22 |

Table 18:   Results for different LLMs under benchmark with and without human filtering. The definitions of abbreviation are identical with Table 1.

**Evaluation result difference before and after human filtering.** We compared the test results of different LLMs under NewTerms 2022, both before human filtering (900 questions) and after human filtering (744 questions). The results are shown in Table 18. We can see that the results under the two benchmark settings are highly consistent. The absolute value of the performance gap between the two settings averages only 1.59 across each different task and each different LLM. Furthermore, the performance ranking among different models remains entirely consistent under both the Base and Gold settings. This proves that our benchmark can achieve the same evaluation abilities and conclusions without human filtering, indicating that human filtering is optional.

Additionally, for the filtered-out questions, the performance of LLMs is slightly higher under the Base setting (+0.95 on average) but lower under the Gold setting (-1.89 on average) compared to the unfiltered questions. This suggests that these filtered-out questions may be biased towards LLMs, making it easier for them to select the auto-generated answers, even though the questions themselves may not be correct.

## D   Main Results on More Open-Sourced LLMs

We also employ the following LLMs for our experiments: Vicuna-1.3 (7B and 13B) [77], fine-tuned from Llama [61]; ChatGLM-2 (6B) [74]; Baichuan-2 (7B and 13B) [69]; Qwen (7B and 14B) [6]; and Mistral (7B) [35]. All tests are done under greedy decoding. Experimental results are shown in Table 19. As indicated by the results, except for Vicuna-1.3, which performed poorly on our tasks and failed to understand the question well, the experimental results of the remaining models all maintain the conclusions obtained in the main text. Among them, the Qwen-Chat model achieved the best results on both Base and Gold, followed by Mistral-Instruct-0.1.

| LLM | Size | NewTerm 2022 | | | | | NewTerm 2023 | | | | |
|---|---|---|---|---|---|---|---|---|---|---|---|
| | | COMA | COST | CSJ | Avg. | Gold | COMA | COST | CSJ | Avg. | Gold |
| Vicuna-1.3 | 7B | 30.46 | 24.78 | 58.94 | 38.06 | 44.20 | 25.88 | 32.77 | 71.85 | 43.50 | 44.49 |
| | 13B | 30.59 | 23.91 | 65.77 | 40.09 | 43.73 | 25.88 | 32.34 | 80.08 | 46.10 | 50.02 |
| ChatGLM-2 | 6B | 42.09 | 43.77 | 51.99 | 45.95 | 64.43 | 31.87 | 60.17 | 56.31 | 49.45 | 62.07 |
| Baichuan-2-Chat | 7B | 40.00 | 42.90 | 63.06 | 48.65 | 72.90 | 48.25 | 58.05 | 79.02 | 61.77 | 74.72 |
| | 13B | 41.44 | 50.72 | 60.10 | 50.76 | 76.88 | 46.78 | 64.55 | 64.67 | 58.67 | 76.71 |
| Qwen-Chat | 7B | 44.31 | 50.43 | 68.08 | 54.28 | 83.95 | 46.35 | 65.11 | 83.53 | 65.00 | 85.22 |
| | 14B | 50.85 | 49.13 | 68.73 | 56.24 | 87.14 | 56.43 | 65.54 | 83.13 | 68.37 | 90.10 |
| Mistral-Instruct-0.1 | 7B | 43.53 | 42.61 | 56.76 | 47.63 | 79.25 | 44.44 | 57.49 | 66.80 | 56.24 | 80.31 |
| Average | - | 40.41 | 41.03 | 61.68 | 47.71 | 69.06 | 40.75 | 54.50 | 73.17 | 56.14 | 70.46 |

Table 19:  Results for more different LLMs on NewTerm 2022 and 2023. The order of the LLMs is based on their release date in HuggingFace, with the earliest at the top. The definitions of the abbreviations are the same as in Table 1.

# E    Case Study for LLMs of Different Year

We also present specific examples that illustrate the differences in how earlier models and more recent models interpret new terms, highlighting the advancements made by newer models in understanding recent or domain-specific vocabulary. To further explore this, we analyzed cases involving Llama-2-Chat-70B and Llama-3-Instruct-70B, focusing on concepts that earlier LLMs overlooked but more recent models successfully identified.

- **New Term:** *supercloud*
- **Meaning:** Noun, a single computing system where services such as storage, apps, etc. from different providers can be easily accessed by the user.
- **Question:** Businesses are adopting ***superclouds*** to streamline integration across various digital service platforms. Is this example in line with commonsense and grammatically correct?
- **Llama-2 Response:** Incorrect (X)
- **Llama-3 Response:** Correct (✓)
- **Llama-2 Meaning Guessing:** A supercloud is a massive, powerful cloud that is formed by the combination of several smaller clouds, suggesting a large and potentially threatening weather system.
- **Llama-3 Meaning Guessing:** The word "supercloud" likely refers to an extremely large or powerful cloud, either in a literal sense (e.g., a massive storm cloud) or a figurative sense (e.g., a vast and dominant cloud computing platform).

In response to our question containing the new term "supercloud," under the zero-shot Base setting, Llama-2 incorrectly labeled this as "Incorrect," whereas Llama-3 accurately classified it as "Correct." To further investigate, we analyzed how each model interpreted the meaning of the term. We found that Llama-2 solely associated the term with meteorological contexts, while Llama-3 correctly connected it to cloud computing. This difference highlights the older model's limitations and misjudgments due to its incomplete grasp of newer technological terms.

Additionally, we present another case study that explores different types of new terms and tasks:

- **New Term:** *stochastic parrot*
- **Meaning:** Noun, a way of describing a large language model, because it can produce text that sounds natural but does not understand what it is saying.
- **Question:** The _ flawlessly recites poetry without grasping the underlying emotions. In the previous sentence, does _ refer to A. ***Stochastic parrot***, B. Aware person, C. Probabilistic repeater, or D. Stocky patriot?
- **Llama-2 Response:** C (X)
- **Llama-3 Response:** A (✓)
- **Llama-2 Meaning Guessing:** A stochastic parrot is a parrot that engages in random and unpredictable behavior, possibly due to its exposure to certain environmental factors or its natural temperament.
- **Llama-3 Meaning Guessing:** The term "stochastic parrot" likely refers to a machine learning model or artificial intelligence that generates responses or outputs in a seemingly random or unpredictable manner, much like a parrot mimicking sounds, but with a nod to the mathematical concept of stochasticity, implying a probabilistic or chance-based process.

This case demonstrates that Llama-2 perceived the term "stochastic parrot" in its literal sense, leading to a misinterpretation of the task, while Llama-3 accurately recognized its metaphorical usage to describe an AI's capabilities, correctly guiding its response to the question.

# F Benchmark Construction with Different LLMs

In the main text, we primarily used `gpt-4-0613` to generate the benchmark. It is worth noting that although our benchmark generation process benefits from stronger LLMs, it does not rely on any specific LLM. As for the universality of our pipeline with other LLMs, we constructed a new benchmark using Claude, i.e., `claude-2.1`, based on the 300 new words we collected in 2022. We also employed the same construction framework and filtering methods. Finally, before human filtering, we obtained 900 questions, aligning with the generation of NewTerm 2022.

| | NewTerm 2022 with GPT-4 | | | | NewTerm 2022 with Claude-2.1 | | | |
|---|---|---|---|---|---|---|---|---|
| | Multi. | Zero | Wrong | Acc. (%) | Multi. | Zero | Wrong | Acc. (%) |
| COMA | 49 | 54 | 97 | 77.78 | 81 | 51 | 176 | 65.78 |
| COST | 50 | 55 | 30 | 85.00 | 36 | 136 | 38 | 76.67 |
| CSJ | - | - | 140 | 84.44 | - | - | 121 | 86.56 |

Table 20: The number of cases where the automatically generated answer does not align with human annotation. The abbreviations are the same as defined in Table 17.

Subsequently, we adopted the same human filtering approach as the main text. We calculate the inter-annotator agreement using Fleiss' Kappa, which reaches a score of 0.67. Additionally, in 76.33% of cases, the annotator results match the automatically generated ones. These results are slightly lower than those of GPT-4 (0.70 / 82.41%), but still comparable. Detailed analysis of error reasons is given in Table 20. For the COMA task, which requires a multi-step generation process, the error rate is more significantly affected by the LLM's capabilities. However, for tasks that only require one or two steps of generation, such as CSJ, the impact is smaller.

We further analyze the performance of different LLMs under NewTerm 2022, with experimental settings aligned with those in the main text in Section 4.1. The results are shown in Table 21. The performance ranking of different LLMs and the performance changes under different settings are consistent with NewTerm 2022 generated by `gpt-4-0613`, demonstrating the effectiveness of using different LLMs to generate benchmarks.

| LLM | Size | Base | | | | Gold | | | |
|---|---|---|---|---|---|---|---|---|---|
| | | COMA | COST | CSJ | Avg. | COMA | COST | CSJ | Avg. |
| Vicuna-1.3 | 7B | 31.07 | 26.50 | 57.92 | 38.49 | 32.85 | 29.43 | 69.50 | 43.92 |
| | 13B | 35.71 | 27.75 | 61.00 | 41.49 | 31.88 | 29.15 | 86.87 | 49.30 |
| ChatGLM-2 | 6B | 46.44 | 46.44 | 46.46 | 46.45 | 74.92 | 73.92 | 46.98 | 65.27 |
| Llama-2-Chat | 7B | 31.55 | 25.80 | 78.12 | 45.16 | 55.34 | 61.09 | 89.96 | 68.80 |
| | 13B | 46.12 | 34.45 | 33.72 | 38.10 | 72.98 | 54.53 | 48.91 | 58.81 |
| | 70B | 56.15 | 43.65 | 41.96 | 47.25 | 88.35 | 78.94 | 74.77 | 80.69 |
| Baichuan-2 | 7B | 52.27 | 44.49 | 72.97 | 56.58 | 77.51 | 74.06 | 64.99 | 72.19 |
| | 13B | 57.61 | 45.75 | 55.73 | 53.03 | 79.94 | 76.57 | 66.02 | 74.18 |
| Qwen | 7B | 53.07 | 47.70 | 63.58 | 54.78 | 80.58 | 76.29 | 77.48 | 78.12 |
| | 14B | 57.77 | 45.19 | 74.13 | 59.03 | 88.03 | 90.10 | 89.19 | 89.10 |
| Mistral | 7B | 49.68 | 44.77 | 44.92 | 46.45 | 84.63 | 77.27 | 57.53 | 73.14 |
| Llama-3-Instruct | 8B | 61.49 | 47.56 | 52.64 | 53.90 | 91.59 | 89.12 | 90.09 | 90.27 |
| | 70B | 68.12 | 60.11 | 53.02 | 60.42 | 95.31 | 96.37 | 88.55 | 93.41 |
| GPT-3.5-0613 | - | 61.00 | 48.54 | 67.44 | 58.99 | 88.35 | 91.21 | 89.96 | 89.84 |
| GPT-4-0613 | - | 71.04 | 60.81 | 59.72 | 63.85 | 94.98 | 97.07 | 91.63 | 94.56 |
| Average | - | 51.94 | 43.30 | 57.56 | 50.93 | 75.82 | 73.01 | 75.50 | 74.77 |

Table 21: Results for different LLMs on benchmark generated by `claude-2.1` based on terms from 2022. The order of the LLMs is based on their release date in HuggingFace, with the earliest at the top, except for GPT series models. The definitions of abbreviation are identical with Table 1.

# G   Datasheet for NewTerm

In this section, we provide more detailed documentation of the dataset with the intended uses. We base ourselves on the datasheet proposed by Gebru et al. [24].

## G.1   Motivation

**For what purpose was the dataset created?**   The NewTerm benchmark focuses on the real-time evaluation of LLMs, which is crucial for their effectiveness. Specifically, we concentrate on the less-explored area of new term evaluation and propose a highly automated benchmark construction pipeline to ensure real-time updates and generalization to a wider variety of terms. Our ultimate goal is to develop an efficient benchmark for tracking LLMs' ability to understand new terms, and we will update it annually. Furthermore, we can also assess the performance of different LLMs and potential improvement strategies.

**Who created the dataset (e.g., which team, research group) and on behalf of which entity (e.g., company, institution, organization)?**   The NewTerm benchmark was developed with contributions from the authors of this paper and was supported by the Institute of Computing and Intelligence at Harbin Institute of Technology, Shenzhen, China.

**Who funded the creation of the dataset?**   The dataset was funded by multiple grants, as detailed in the acknowledgments section.

## G.2   Composition

**What do the instances that comprise the dataset represent (e.g., documents, photos, people, countries)?**   Each instance consists of a question covering three tasks, introduced in Section 3.3. These questions are generated in a highly automated manner by our construction pipeline.

**How many instances are there in total (of each type, if appropriate)?**   The benchmark currently consists of 744 questions for NewTerm 2022, and 715 for NewTerm 2023, evaluating the performance of LLMs under in total 600 new terms. We will update the benchmark annually to evaluate the latest year's new terms.

**Does the dataset contain all possible instances or is it a sample (not necessarily random) of instances from a larger set?**   The NewTerm benchmark is a sample of instances from a larger set, where the large set corresponds to the benchmark composed of questions for all new terms collected annually in online dictionaries. We select the most representative 300 new terms from the full set of updated terms each year, covering new words, new phrases, and old words with new meanings, and construct benchmarks for these new terms. This sample covers the most challenging part of the annual new term updates and serves as a typical representation of the full set. For a detailed analysis, please refer to Section 4.4.

**What data does each instance consist of?**   For all tasks, each instance is given in JSON format, including the evaluated "new term", its "meaning", and its "type" by this question. Here, new words correspond to the type "new words not deduced", new phrases correspond to "new phrases not deduced", and old words with new meanings correspond to "old words not deduced". Additionally, it includes a "question", two or four "choices", and the correct answer "gold", which represents the index of the correct choice. For COMA, we additionally include a "split" attribute, indicating whether the selected choice is the cause or the effect of the question. This will correspond to different testing prompts. Below is an example from the COMA task:

```json
{
    "term": "Juggers",
    "meaning": "When the sleeves of a shirt are uncomfortably
        short.",
    "type": "new words not deduced",
    "question": "Several people have started complaining about
        their new Juggers.",
```

```
    "choices": [
        "the company had used low-quality materials, leading to
            rapid wear and tear, much to the customers'
            disappointment and dissatisfaction.",
        "the company failed to clearly communicate the product'
            s dimensions, leading to widespread frustration
            among their customer base.",
        "the fabric quality was sub-par, colors faded after a
            few washes, and sizes were not accurately
            represented on the website.",
        "the trend of body-hugging shirts has led to a spate of
            situations where people ended up with sleeves
            shorter than preferred."
    ],
    "gold": 3,
    "split": "cause"
}
```

**Is there a label or target associated with each instance?**   Yes, as mentioned in the previous question, each instance includes a "gold" field, which corresponds to the index of the correct answer choice.

**Is any information missing from individual instances?**   No, all the instances should have complete information corresponding to the content as well as to the attributes.

**Are relationships between individual instances made explicit (e.g., users' movie ratings, social network links)?**   For each selected new term, we construct multiple instances covering various tasks to evaluate LLMs' understanding ability. To make this relationship explicit, we can match the "term" and "meaning" fields in the instances. Instances with identical term and meaning fields indicate that they are evaluating the same new term.

**Are there recommended data splits (e.g., training, development/validation, testing)?**   The NewTerm benchmark primarily focuses on evaluation, and all instances are part of the test set. For training, we recommend using only the "term" and "meaning" fields in each instance. A clean dataset containing only these two fields is also released and can be directly accessed.

**Are there any errors, sources of noise, or redundancies in the dataset?**   Before human filtering, the NewTerm benchmark contains errors and sources of noise, which are analyzed in detail in Appendix C.1. After human filtering, we effectively removed these errors and noise. There are no redundancies in our benchmark.

**Is the dataset self-contained, or does it link to or otherwise rely on external resources (e.g., websites, tweets, other datasets)?**   The NewTerm benchmark is self-contained.

**Does the dataset contain data that might be considered confidential (e.g., data that is protected by legal privilege or by doctor–patient confidentiality, data that includes the content of individuals' non-public communications)?**   No.

**Does the dataset contain data that, if viewed directly, might be offensive, insulting, threatening, or might otherwise cause anxiety?**   No.

### G.3 Collection Process

**How was the data associated with each instance acquired?**    We initially collected new terms from online dictionaries, including Cambridge[2], Collins[3], and Oxford[4]. Subsequently, the NewTerm benchmark was indirectly derived from other data using our automated framework, as detailed in Section 3.4. We validated and filtered the generated data through human filtering and thorough analysis, as described in Section 3.5.

**What mechanisms or procedures were used to collect the data (e.g., hardware apparatuses or sensors, manual human curation, software programs, software APIs)?**    We downloaded the HTML of online dictionary update pages and extracted new terms and their meanings, which typically correspond to fixed fields. Due to the neat and noise-free format of the dictionaries, we did not need to perform further filtering or validation.

**If the dataset is a sample from a larger set, what was the sampling strategy (e.g., deterministic, probabilistic with specific sampling probabilities)?**    The sampling method is described in Section 3.2, and its further verification can be found in Section 4.4.

**Who was involved in the data collection process (e.g., students, crowdworkers, contractors) and how were they compensated (e.g., how much were crowdworkers paid)?**    Please refer to Appendix C.1.

**Over what timeframe was the data collected?**    Our benchmark is related to the new terms of each year. Currently, NewTerm 2022 covers new terms from January 2022 to March 2023, and NewTerm 2023 covers April 2023 to March 2024. Additionally, we plan to update the benchmark annually, covering new terms from April of each year to March of the following year.

**Were any ethical review processes conducted (e.g., by an institutional review board)?**    N/A.

### G.4 Preprocessing/cleaning/labeling

**Was any preprocessing/cleaning/labeling of the data done (e.g., discretization or bucketing, tokenization, part-of-speech tagging, SIFT feature extraction, removal of instances, processing of missing values)?**    Yes. See Section 3.5.

**Was the "raw" data saved in addition to the preprocessed/cleaned/labeled data (e.g., to support unanticipated future uses)?**    Yes. Both the raw and filtered datasets have been released and can be accessed at `https://github.com/hexuandeng/NewTerm`. The filtered datasets are distinguished by the suffix "_clean".

**Is the software that was used to preprocess/clean/label the data available?**    Yes. We have released the automatic pipeline code for LLM filtering, along with all corresponding frontend and backend codes required for human filtering. These can be accessed at the above url.

### G.5 Uses

**Has the dataset been used for any tasks already?**    Yes. In our submitted paper, we conducted extensive evaluation and comprehensive analysis on numerous versions of mainstream LLMs, aiming to evaluate their performance when facing new terms, as detailed in Section 4.

**Is there a repository that links to any or all papers or systems that use the dataset?**    N/A.

---

[2]`https://dictionaryblog.cambridge.org/category/new-words`
[3]`https://www.collinsdictionary.com/submissions/latest`
[4]`https://www.oed.com/information/updates`

**What (other) tasks could the dataset be used for?** The NewTerm benchmark can also be used for evaluating the performance of LLMs on various other terms beyond new ones, such as religious, literary, and low-frequency terms. To facilitate this, we have released the code for automatic benchmark construction and the human interactive interface construction. This enables developers interested in building their benchmarks for other new terms to do so with ease. Our construction solution is cost-effective, especially when the human filtering step is omitted, making it accessible for developers to build their own benchmarks. We hope this contribution will encourage further research on the performance of different types of terms within the research community.

**Is there anything about the composition of the dataset or the way it was collected and preprocessed/cleaned/labeled that might impact future uses?** No.

**Are there tasks for which the dataset should not be used?** No.

## G.6 Distribution

**Will the dataset be distributed to third parties outside of the entity (e.g., company, institution, organization) on behalf of which the dataset was created?** Yes, the dataset is of public access.

**How will the dataset will be distributed (e.g., tarball on website, API, GitHub)?** The NewTerm benchmark will be made public on a GitHub repository, which can be found at `https://github.com/hexuandeng/NewTerm`. The public content includes the following three parts:

- NewTerm benchmark: Currently, it covers NewTerm 2022 and NewTerm 2023, constructed from new terms in 2022 and 2023, and will continue to be updated annually.

- Testing code: We have released easy-to-use testing code and corresponding instructions, allowing testing on most open-source/closed-source LLMs with just a few commands. For testing other LLMs, we provide detailed guidance, enabling developers to modify minimal code to test their LLMs. Finally, all results in this paper are consistent with the testing framework, ensuring the reproducibility of the reported results.

- Benchmark construction code: We have released the code for automatic benchmark construction and human interactive interface. This supports developers interested in building their benchmarks for other new terms, e.g., religious, literary, and low-frequency terms.

**When will the dataset be distributed?** The NewTerm benchmark is currently available in the GitHub repository referenced in the previous response.

**Will the dataset be distributed under a copyright or other intellectual property (IP) license, and/or under applicable terms of use (ToU)?** The NewTerm benchmark is distributed under a Creative Commons Attribution 4.0 International license (CC BY 4.0).

**Have any third parties imposed IP-based or other restrictions on the data associated with the instances?** No.

**Do any export controls or other regulatory restrictions apply to the dataset or to individual instances?** No.

## G.7 Maintenance

**Who will be supporting/hosting/maintaining the dataset?** The maintenance and extension of NewTerm will be carried out by the authors of the paper.

**How can the owner/curator/manager of the dataset be contacted (e.g., email address)?** For inquiries, please contact hxuandeng@gmail.com.

**Is there an erratum?** No.

**Will the dataset be updated (e.g., to correct labeling errors, add new instances, delete instances)?**
Yes, we will update the benchmark annually to evaluate the performance of the newest LLMs under new terms from the most recent year, covering the period from April of the current year to March of the following year. The authors of this paper will collect these new terms, construct the updated benchmark, and release it on the GitHub repository mentioned in the previous question.

**If the dataset relates to people, are there applicable limits on the retention of the data associated with the instances (e.g., were the individuals in question told that their data would be retained for a fixed period of time and then deleted)?**  N/A.

**Will older versions of the dataset continue to be supported/hosted/maintained?**  Yes, we will continue to support, host, and maintain older versions of the dataset in the open-source repository. This will enable tracking the performance of LLMs over time as terms evolve.

**If others want to extend/augment/build on/contribute to the dataset, is there a mechanism for them to do so?**  Yes. We have released the code for automatic benchmark construction and the human interactive interface construction, which supports developers interested in building their benchmarks for other new terms. Contributors can use these codes to generate datasets for model evaluation or improvement. The new datasets can be distributed independently by the contributors themselves, or they can contact the authors of this paper via email. We will manually review them and decide whether to publish them in the GitHub repository.

### G.8   Further Statement

- The authors of the paper bear all responsibility in case of violation of rights, etc., and confirmation of the data license. We confirm the use of the CC BY 4.0 license for the data.

- We ensure that all results are easily reproducible in Appendix G.6, guarantee that all results can be easily reproduced, i.e. all necessary datasets, code, and evaluation procedures are accessible and documented in our GitHub repository.

- We release the NewTerm benchmark along with the associated construction and evaluation code at `https://github.com/hexuandeng/NewTerm`, ensuring that the dataset will be available for a long time. We will continue hosting and maintaining this benchmark, updating it annually with the latest year's data to support tracking the real-time abilities of LLMs. The dataset format is in JSONL.

- To ensure our benchmark can be discovered and organized by anyone, we publish it on Hugging Face at `https://huggingface.co/datasets/hexuandeng/NewTerm`, which will automatically add structured metadata to the dataset.

