# OpenReview forum: "NewTerm: Benchmarking Real-Time New Terms for Large Language Models with Annual Updates"
_NeurIPS.cc/2024/Datasets_and_Benchmarks_Track — NeurIPS 2024 Track Datasets and Benchmarks Poster_

### Official Review · Reviewer_LL5Y · 2024-06-20

**Rating:** 7
**Confidence:** 3

**Review:**

- **Originality**: The paper is original in my opinion. To my knowledge, it is among the first works to study how LLMs perform on new terms beyond their knowledge cutoff. It also presents a systematic data curation pipeline that builds reliable, indicative questions from these terms.

- **Clarity**: This paper is mostly clearly written. However, some minor parts of this paper are confusing. Please see "opportunities for improvements".

- **Significance**. It is known that LLMs have knowledge cutoffs, and that it is costly to update LLMs with the latest knowledge. However, no systematic analysis on this phenomenon is performed. From this perspective, I appreciate the authors efforts in providing such a systematic analysis. The data curation pipeline is also valuable for researchers to create other benchmarks.

- **Quality**. To me, the methodology adopted in this paper are mostly sound, including the data filtering as well as the analytical part.

**Strengths:**

1. **Systematic and Rigorous of New Terms in the Context of LLMs. ** The analysis in this paper is systematic in my opinion. It applies strict time cutoffs, not only in the terms, but also in the evaluated LLMs. Also, the curation of the benchmark is also systematic, which I appreciate. I appreciate the careful selection of new terms/words with new meanings in Section 3.2 according to frequency and deducing difficulty. I also appreciate the authors' efforts in generating meaningful negative choices with 'related terms' (while to my knowledge, existing benchmarks do not generally care about negative examples). Therefore,  I think that the systematic and rigorous benchmark curation is a notable strength of this paper.
2. **Evaluation is also carefully done**. I appreciate the authors' efforts in not only presenting the accuracy, but also presenting a baseline (upper bound), i.e. the "Gold" Accuracy. While a mistake can be caused by either not knowing the new term, or by being unable to reason, comparing with the "Gold" accuracy rules out the second reason (i.e. unable to reason for the correct answer), which makes the analysis more pertinent to the goal of this paper.
3. **Highly Automated Method for Benchmark Generation**. The curation of this benchmark involves minimal human interference. Only a final human annotation is needed, and the authors also demonstrate that the human annotation is actually not necessary (in terms of providing benchmarking results). This automatic pipeline would be helpful to future researchers in curating further time-related benchmarks or LLM benchmarks as a whole.
4. **Paper is well-organized**. The paper is generally well-written and clearly organized.

**Additional Feedback:**

No additional feedback. Please see the above sections.

**Clarity:**

This paper is mostly well written but unclear at times. Please see the 'opportunities for improvements'.

**Correctness:**

I appreciate the rigor used in creating the benchmark and evaluating models, as stated in “Strengths”.

**Documentation:**

The dataset is available at anonymous github. I took a look at the repo and I found it satisfactory with sufficient code, data, and documents.

**Ethics:**

No ethical concerns.

**Limitations:**

No potential negative societal impacts. Limitations are discussed.

**Opportunities For Improvement:**

1. **How to improve LLMs on new terms?** The paper leaves one question unanswered, namely how to improve the performance of LLMs upon new terms. What I can think of is some kind of RAG methods that include webpages about latest trends as the database, and see how RAG-empowered LLMs perform. Although this may be beyond the scope of this paper, I am interested in seeing how it performs.
2. **Why unfiltered samples are used for Section 4.3**. I wonder why unfiltered samples are used in Section 4.3 instead of filtered samples, and the authors did not provide justifications (intuitively, unfiltered samples lead to more noise).
3. **Minor confusions**. I list my confusions during the reading of this paper as follows.
- Section 3.5 Line 222: 'We retain one question per term with lower perplexity calculated by LLaMA-2-7B'. I wonder why lower perplexity leads to more difficult questions?
- Section 4.2 Line 314: "there are 50% more new terms learned in 2022 compared to 2023... LLMs can update new terms within their knowledge cutoff but are hard to generalize to terms from more distant periods." I must be missing something, but why does 'learn more from 2022' lead to 'hard to generalize to terms from more distant periods'?
- Section 4.3 Line 334, 'in COMA and COST, 'Frequent' terms also tend to perform poorly'. Is there any reason behind that? Intuitively, frequent words on the web appear more in the training data, so this observation seems odd. Can it be related to data noise?

**Relation To Prior Work:**

This paper is original in my opinion.

**Summary And Contributions:**

This paper presents a dataset and benchmark, NewTerms, for evaluating the ability of LLMs to understand new terms. The NewTerm benchmark is constructed with a (mostly) automatic pipeline, and is carefully curated to ensure its credibility. With NewTerm, the authors perform benchmarking studies on popular LLMs and come up with helpful insights on how LLMs learn new terms.

---

> ### Author Response · Authors · 2024-08-24
> **Response to Reviewer LL5Y (2/2)**
>
> > Q4: Section 4.2 Line 314: "there are 50% more new terms learned in 2022 compared to 2023... LLMs can update new terms within their knowledge cutoff but are hard to generalize to terms from more distant periods." I must be missing something, but why does 'learn more from 2022' lead to 'hard to generalize to terms from more distant periods'?
>
> The key point here is that the generalization ability of models for learning new terms is quite limited. Models struggle to adapt to new terms they haven't encountered before, especially those from distant periods that exceed the knowledge cutoff. This conclusion is drawn from the following observation: newer models, such as Llama-3 and GPT-4-0125, have a knowledge cutoff (December 2023) that covers all NewTerm 2022 time points (January 2022 to March 2023) and approximately two-thirds of the NewTerm 2023 time points (April 2023 to March 2024). Our statistics show that the number of learned terms over these two years follows a 3:2 ratio, which is roughly proportional to the time span each model covers. This suggests that models have difficulty understanding new terms even within a short period following the knowledge cutoff. This shows that terms beyond the knowledge cutoff are difficult for models, requiring manual intervention.
>
>
> > Q5: Section 4.3 Line 334, 'in COMA and COST, 'Frequent' terms also tend to perform poorly'. Is there any reason behind that? Intuitively, frequent words on the web appear more in the training data, so this observation seems odd. Can it be related to data noise?
>
> The term "Frequent" refers to one of the dimensions we use for classifying new terms. It represents those terms with the highest frequency among the most recently updated terms across dictionaries. Since our selection is based on newly updated terms rather than all terms, this category does not simply represent high-frequency terms. Instead, it highlights high-frequency terms that have recently acquired new meanings—old terms with new meanings. Such terms have been shown to be particularly challenging in our experiments, e.g., in Figure 12. Intuitively, because the model is more familiar with the term's previous meanings, this can interfere with its comprehension of the new meaning, sometimes leading to a more pronounced performance decline than with entirely new terms.

---

> ### Author Response · Authors · 2024-08-24
> **Response to Reviewer LL5Y (1/2)**
>
> We appreciate the valuable comments from the reviewers and respond as below.
>
>
> > Q1: How to improve LLMs on new terms? The paper leaves one question unanswered, namely how to improve the performance of LLMs upon new terms. What I can think of is some kind of RAG methods that include webpages about latest trends as the database, and see how RAG-empowered LLMs perform. Although this may be beyond the scope of this paper, I am interested in seeing how it performs.
>
> Thank you for your suggestion. Your advice is instrumental in helping us develop a more comprehensive research framework. We have included supplementary experiments focused on RAG improvements. For details on the experimental settings and results, please refer to Q2 of the General Response.
>
>
> > Q2: Why unfiltered samples are used for Section 4.3. I wonder why unfiltered samples are used in Section 4.3 instead of filtered samples, and the authors did not provide justifications (intuitively, unfiltered samples lead to more noise).
>
> We generated a distinct and more comprehensive dataset in Section 4.3 in order to minimize the randomness when evaluating LLMs. Specifically, this dataset incorporates a broader range of new terms and is significantly larger in scale compared to the main text. The main text focused on a filtered set of new terms, comprising new words, new phrases, and old words with new meanings, totaling 300 terms annually. In this section, to expand our investigation, we have revisited some of the new terms that were initially excluded during the "New Term Selection" process. Consequently, we have increased the total count of new terms to 1,200 and generated a more comprehensive dataset comprising approximately 20,000 questions to minimize randomness. This new dataset's large scale increases the cost and complexity of manual filtering. Furthermore, our previous work has demonstrated that unfiltered datasets can still provide accurate system-level comparisons, which justifies our choice to use an unfiltered dataset in this instance.
>
>
> > Q3: Section 3.5 Line 222: 'We retain one question per term with lower perplexity calculated by LLaMA-2-7B'. I wonder why lower perplexity leads to more difficult questions?
>
> Thank you for pointing that out. We apologize for the typo; it should be "higher perplexity." Additionally, to verify the effectiveness of our selection scheme, we conducted experiments comparing several metrics. However, to avoid an overly lengthy appendix, these were not included. Below are the metrics we used to evaluate the difficulty of questions for each term, along with the experimental results:
>
> Metrics: To select more challenging questions for each term $w$, we construct metrics to evaluate the difficulty among $N$ questions. Specifically, we define four different difficulty metrics, i.e., 1) the perplexity of the question $\textrm{PPL}(s_{i})$ (**PPL**), 2) the perplexicity without new terms $\textrm{PPL}(s_{i}-w)$ (**Re. \& PPL**), 3) Shapley Value $\textrm{PPL}(s_{i})-\textrm{PPL}(s_{i}-w)$ (**Shapley**), and 4) Sentence length (**Length**). Here, $s_{i}$ is the corresponding complete sentence of the question, and $s_{i}-w$ means replacing the new term $w$ in $\mathbf{s}_{i}$ with <$pad$>. Perplexity is calculated by Llama-2-7B.
>
> Results for each metric: To test which metric selects the most difficult questions, we construct experiments under the benchmark constructed in Section 4.3. We select one question per term with the *highest* score with each metric and calculate the average score of the selected subset. Results are shown in the following table. The PPL metric consistently offers the best results, selecting a subset with lower accuracy than the average score. This justifies our use of PPL as the metric in the "Question Selection" procedure.
>
> |			| PPL 		| Re. & PPL 	| Shapley	| Length	|
> |:--------------------------|:----------------|:-----------------|:----------------|:-----------------|
> | COMA		| 65.77		| 68.30		| 70.09		| 68.39		|
> | COST		| 58.80		| 71.73		| 73.64		| 65.88		|
> | CSJ			| 69.23		| 70.93		| 71.35		| 79.09		|

---

> > ### Comment · Reviewer_LL5Y · 2024-08-26
> > **Rebuttal Acknowledged**
> >
> > I have read the author rebuttal. They help understand the paper better. I will vote to accept this paper.

---

### Official Review · Reviewer_HEp8 · 2024-07-05

**Rating:** 7
**Confidence:** 4
**Correctness:** Yes
**Clarity:** Yes

**Review:**

- I wonder if the authors could consider a more fine-grained granularity, say monthly or quarterly update. LLM research and development is moving very fast, while "annual" updates might be a bit slow. For instance, the experiments in this work are only up to "NewTerm 2023", while we are already half a year into 2024 and many of the LLMs have already seen 2024 data.

- In addition to just prompting various LLMs for evaluation, maybe some of the improvement strategies could also be conducted. For instance, what would be the impact of continued pretraining and retrieval augmentation on NewTerm? I would say retrieval-augmented generation is the most common strategy to bring LLMs back to speed with current affairs, so it would be nice to somehow reflect this in the paper.

- In addition to just reporting performance numbers, it would be nice to dive a bit deeper and present some analysis. For example, I would suspect the picking up of new terms would involve cultural implications: LLMs might be picking up more from Western cultures and contexts while struggle to be equitable towards terms and concepts across diverse cultures, language speakers, and communities. Other elements and/or confounders could also be explored.

- It would be nice to present some concrete examples in the main paper: for example, concepts where earlier/certain families of LLMs overlooked but more recent/more multilingual LLMs successfully captured. In addition, there are several tasks associated with each concept, maybe for some terms LLMs could do well in one task but not the other? This would add more insights to this work in addition to a lot of numbers.

**Strengths:**

+ evolving benchmark is an important research question
+ the automatic update of NewsTerm is promising

**Additional Feedback:**

please see above

**Documentation:**

Yes

**Limitations:**

Yes

**Opportunities For Improvement:**

please see above

**Relation To Prior Work:**

Yes

**Summary And Contributions:**

This work presents NewTerm, an evolving QA benchmark to test LLMs' knowledge of emerging terms in dictionaries. Experiments demonstrate that open and closed LLMs are not doing so well on this front, while the automatic construction and periodic update of this benchmark is promising.

---

> ### Author Response · Authors · 2024-08-24
> **Response to Reviewer HEp8 (3/3)**
>
> > Q5: In addition, there are several tasks associated with each concept, maybe for some terms LLMs could do well in one task but not the other? This would add more insights to this work in addition to a lot of numbers.
>
> Thank you for your suggestion. This aspect of the case will greatly help us illustrate the differences in focus between various tasks. In the zero-shot Base setting of Llama-3-Instruct-70B, we examined the new term "wokely," defined as an adjective meaning "Of little worth; poor, mean, paltry." The model's performance varied across three tasks:
>
> | Task		| Question	| Response |
> |:--------------------------|:----------------|:-----------------|
> |COMA| The book's cover was described as wokely by several reviewers. I am hesitating among these options. Help me choose the more likely effect: A. it struggled to attract attention on the bookstore displays despite a compelling narrative inside. B. many readers were enticed to buy it, strengthening its presence on the bestseller list. C. readers were intrigued and the book's sales experienced an unexpected surge worldwide. D. the publisher decided to release a limited edition with a special hardback velvet cover.|A (√)|
> |COST| The goods at the flea market appeared distinctly _, making it hard to find a satisfying purchase. In the previous sentence, does _ refer to A. Spokely, B. Cokely, C. Wokely, or D. Worthy?|D (X)|
> |CSJ| His contributions to the project were considered wokely, barely making any impact. Is this example in line with commonsense and grammatically correct?| Incorrect (X)|
>
> As observed, the model only answered correctly in the COMA task, while failed in the other two tasks. In COMA, the model successfully inferred that "wokely" carries a negative connotation, allowing it to correctly choose choice A. This demonstrates its ability to *comprehend* the new term within a *helpful context*. However, in the COST task, where the model needed to *utilize* new terms and *distinguish* from similar choices, it struggled. Although the phrase "hard to find a satisfying purchase" suggested the need for a negative term, the model incorrectly chose "Worthy," which is grammatically correct but semantically incorrect. In the CSJ task, the model was required to *process* and *interpret* new terms in the *absence* of helpful *context*. The context matched the definition of "wokely" perfectly, yet the model erroneously judged the response as incorrect because it was a judgment-based evaluation.
>
> These results provide a comprehensive evaluation of the model's understanding of the term "wokely." They reveal that while the model can recognize that it is a negative term when the context is clear, it struggles to grasp the detailed meaning of the term and how to accurately use it in different contexts.

---

> > ### Comment · Reviewer_HEp8 · 2024-08-26
> >
> > I would like to thank the authors for their detailed response: I adjusted the rating accordingly.

---

> ### Author Response · Authors · 2024-08-24
> **Response to Reviewer HEp8 (2/3)**
>
> > Q4: It would be nice to present some concrete examples in the main paper: for example, concepts where earlier/certain families of LLMs overlooked but more recent/more multilingual LLMs successfully captured.
>
> Thank you for your suggestion; this advice is instrumental in helping us uncover and present more intuitive and effective insights. For concepts that earlier LLMs overlooked but more recent models successfully identified, we conducted a further analysis on cases involving Llama-2-Chat-70B and Llama-3-Instruct-70B:
>
> | New Term		| supercloud 	|
> |:--------------------------|:----------------|
> |Meaning	| noun, a single computing system where services such as storage, apps, etc. from different providers can be easily accessed by the user.		|
> |Question	| Businesses are adopting superclouds to streamline integration across various digital service platforms. Is this example in line with commonsense and grammatically correct?	|
> |Llama-2 Response	| Incorrect  (X)		|
> |Llama-3 Response 	| Correct (√)		|
> |Llama-2 Meaning Guessing	| A supercloud is a massive, powerful cloud that is formed by the combination of several smaller clouds, suggesting a large and potentially threatening weather system.|
> |Llama-3 Meaning Guessing	| The word "supercloud" likely refers to an extremely large or powerful cloud, either in a literal sense (e.g., a massive storm cloud) or a figurative sense (e.g., a vast and dominant cloud computing platform).|
>
> In response to our question containing the new term “supercloud”, under the zero-shot Base setting, Llama-2 incorrectly labeled this as "Incorrect," whereas Llama-3 accurately classified it as "Correct." To further investigate, we analyzed how each model interpreted the meaning of the term. We found that Llama-2 solely associated the term with meteorological contexts, while Llama-3 correctly connected it to cloud computing. This difference highlights the older model's limitations and misjudgments due to its incomplete grasp of newer technological terms.
>
> Additionally, we present another case study that explores different types of new terms and tasks:
>
> | New Term		| stochastic parrot	|
> |:--------------------------|:----------------|
> |Meaning	| noun, a way of describing a large language model, because it can produce text that sounds natural but it does not understand what it is saying.		|
> |Question	| The _ flawlessly recites poetry without grasping the underlying emotions. In the previous sentence, does _ refer to A. Stochastic parrot, B. Aware person, C. Probabilistic repeater, or D. Stocky patriot? |
> |Llama-2 Response	| C (X)	|
> |Llama-3 Response 	| A (√)		|
> |Llama-2 Meaning Guessing	| A stochastic parrot is a parrot that engages in random and unpredictable behavior, possibly due to its exposure to certain environmental factors or its natural temperament.|
> |Llama-3 Meaning Guessing	| The term "stochastic parrot" likely refers to a machine learning model or artificial intelligence that generates responses or outputs in a seemingly random or unpredictable manner, much like a parrot mimicking sounds, but with a nod to the mathematical concept of stochasticity, implying a probabilistic or chance-based process.|
>
> This case demonstrates that Llama-2 perceived the term "stochastic parrot" in its literal sense, leading to a misinterpretation of the task, while Llama-3 accurately recognized its metaphorical usage to describe an AI's capabilities, correctly guiding its response to the question.

---

> ### Author Response · Authors · 2024-08-24
> **Response to Reviewer HEp8 (1/3)**
>
> We appreciate the valuable comments from the reviewers and respond as below.
>
>
> > Q1: I wonder if the authors could consider a more fine-grained granularity, say monthly or quarterly update. LLM research and development is moving very fast, while "annual" updates might be a bit slow. For instance, the experiments in this work are only up to "NewTerm 2023", while we are already half a year into 2024 and many of the LLMs have already seen 2024 data.
>
> Thank you for your suggestion. The primary aim of our benchmark is to track the ongoing knowledge updates of LLMs, aligning our update frequency with that of mainstream LLM models. While LLMs undergo periodic updates, the frequency of their knowledge cutoff updates remains relatively infrequent. For instance, the GPT-4 series has only been updated three times regarding its knowledge cutoff, in September 2021, April 2023, and December 2023. GPT-3.5 has maintained its knowledge cutoff in September 2021 since its launch. Similarly, GPT-4o and GPT-4o mini have consistently had a knowledge cutoff of October 2023 since their introduction.
>
> Additionally, to exclude pre-existing or easily deduced terms, we utilize LLMs with specific knowledge cutoffs: September 2021 for NewTerm 2022, and April 2023 for NewTerm 2023. This necessitates that our updates cannot precede those of the LLMs' knowledge cutoffs. However, faster updates are not required as our goal is to track the LLMs' knowledge updates.
>
> Finally, NewTerm 2023, which includes terms up to March 2024, is not outdated, as it misses only the terms introduced in the last two months prior to the submission deadline. To better accommodate an update frequency of less than one year and to reflect the most recent terms, we will adjust our naming convention: NewTerm 2022 will be renamed to NewTerm 2023.03, and NewTerm 2023 will become NewTerm 2024.03. Since overly detailed updates provide limited benefits for tracking knowledge cutoffs, we plan to update our benchmark either semi-annually or annually, synchronizing our updates with the knowledge cutoff update intervals of mainstream LLMs.
>
>
> > Q2: In addition to just prompting various LLMs for evaluation, maybe some of the improvement strategies could also be conducted. For instance, what would be the impact of continued pretraining and retrieval augmentation on NewTerm? I would say retrieval-augmented generation is the most common strategy to bring LLMs back to speed with current affairs, so it would be nice to somehow reflect this in the paper.
>
> Thank you for your suggestion. Your advice helps us build a more comprehensive research framework. We have added supplementary experiments for RAG improvements. For details on the experimental settings and results, please refer to Q2 of the General Response.
>
>
> > Q3: In addition to just reporting performance numbers, it would be nice to dive a bit deeper and present some analysis. For example, I would suspect the picking up of new terms would involve cultural implications: LLMs might be picking up more from Western cultures and contexts while struggling to be equitable towards terms and concepts across diverse cultures, language speakers, and communities. Other elements and/or confounders could also be explored.
>
> Thank you for your suggestion; this provides an excellent perspective to consider. Using NewBench 2023 as an example, we selected professional terms and regional dialects to analyze their performance variations. For detailed experimental settings and results, please refer to Q1 of the General Response.

---

### Official Review · Reviewer_7eHY · 2024-07-24
**Review of paper 1691**

**Rating:** 7
**Confidence:** 4
**Clarity:** Well written.

**Review:**

This paper makes a innovative contribution to the field of NLP by introducing a benchmark that addresses the real-time adaptation capabilities of LLMs. The high-quality automated construction and comprehensive evaluation approach are commendable. While there are areas for improvement, particularly in data generation quality and reducing dependency on human filtering. This benchmark has the potential to become a crucial tool for evaluating and enhancing LLMs’ performance in dynamic, real-world environments.

**Strengths:**

1. The paper is generally clear and easy to follow.
2. The use of multiple tasks (COMA, COST, and CSJ) provides a thorough assessment of LLMs’ performance with new terms, offering detailed insights into their strengths and weaknesses.
3. The benchmark, along with datasets and code, is openly available, promoting transparency and collaboration within the research community.

**Additional Feedback:**

NA.

**Correctness:**

The claims made in the submission are supported by the data and analysis presented. The NewTerm benchmark is constructed using a methodologically sound approach. The evaluation methods and experiment design appear appropriate and are executed as described.

**Documentation:**

The submission includes sufficient detail on data collection and organization, availability and maintenance, and ethical and responsible use. For the benchmark, the documentation supports reproducibility by providing comprehensive information on the methodological approach, evaluation methods, and availability of code and data.

**Ethics:**

NA.

**Limitations:**

See Opportunities For Improvement

**Opportunities For Improvement:**

1. The benchmark is designed primarily for English language terms. Its applicability to other languages or multilingual models is not addressed, which could limit its relevance for researchers working on non-English or multilingual NLP systems.
2. The benchmark may not fully represent specialized terms from niche domains (e.g., medical, legal, or technical jargon), which could limit its usefulness for evaluating models in those specific areas.
3. The process of selecting new terms from online dictionaries might introduce biases based on the sources and the criteria used for term inclusion. This could affect the representativeness and fairness of the benchmark.

**Relation To Prior Work:**

Yes.

**Summary And Contributions:**

Large language models (LLMs) exhibit significant limitations in handling real-time information due to the knowledge cutoffs in their development. Current benchmarks are outdated and fail to address new terms, posing challenges for real-time updates. To overcome this, the authors introduce NewTerm, an adaptive benchmark for real-time evaluation of new terms. This benchmark employs a highly automated construction method, ensuring high-quality updates with minimal human effort. Empirical results show that LLMs experience over a 20% performance reduction when encountering new terms. Moreover, while updating the knowledge cutoff helps cover some new terms, it does not generalize well to more distant ones. The study also analyzes the types of terms that are more challenging and why LLMs struggle with them, providing insights for future research. The NewTerm benchmark includes data from 2022 and 2023 and will be updated annually.

---

> ### Author Response · Authors · 2024-08-24
> **Response to Reviewer 7eHY**
>
> We appreciate the valuable comments from the reviewers and respond as below.
>
> Your concerns primarily relate to the coverage of new terms in our benchmark, highlighting the absence of terms from non-English languages, specialized domains like law and medicine, and terms not found in online dictionaries.
>
> It is worth noting that our contribution extends beyond merely creating a benchmark for new terms in dictionaries; we’ve developed a highly automated, cost-effective construction pipeline that enables high-quality, system-level LLM evaluation without human intervention. This pipeline requires only the term and its meaning to automatically generate a benchmark covering three related tasks. While our current benchmark focuses on new English terms from dictionaries, the design is highly flexible for expansion. Researchers interested in multilingual terms or specialized domains can efficiently create benchmarks with minimal time and budget. Therefore, even if the benchmark provided in this paper has some coverage limitations, the construction pipeline itself offers substantial value to NLP researchers exploring various term types and provides a reliable test bed for evaluating LLMs' understanding of a wider range of terms.
>
> Below, we will provide further responses to the three questions:
>
> > Q1: The benchmark may not fully represent specialized terms from niche domains (e.g., medical, legal, or technical jargon), which could limit its usefulness for evaluating models in those specific areas.
>
> Thank you for your addition. While it may not encompass all aspects, our benchmark has indeed incorporated a substantial amount of specialized terminology and can effectively generate relevant test questions for these terms. To further underscore the breadth of terms covered and the versatility of our pipeline, we emphasize that the terms sourced from dictionaries encompass not only professional terminology but also dialects. For further details and analysis, please refer to Q1 of our General Response. This demonstrates our pipeline's accuracy in handling a wide range of terms, including those from professional fields and different cultures. Hence, within the existing benchmark, we possess the capability to assess specialized terms originating from a portion of niche domains. As we acquire a broader and more profound lexicon, our pipeline will facilitate the generation of pertinent high-quality questions, offering direction for the evaluation and enhancement of novel terms in their corresponding fields.
>
> > Q2: The process of selecting new terms from online dictionaries might introduce biases based on the sources and the criteria used for term inclusion. This could affect the representativeness and fairness of the benchmark.
>
> At the outset of our research, we considered several data sources, including online forums, search engines, and wikipedia. Among these, we believe that online dictionaries, as one of the most reliable and comprehensive open-source repositories of new terms, accurately reflect current English language trends and encompass nearly all high-quality emerging terms. We have deliberately included a diverse range of new terms—covering new words, phrases, and existing words with new meanings—to minimize potential biases. This diversity aims to ensure comprehensive and unbiased coverage of new terms. Additionally, to address concerns about relying on a single data source, we expanded our benchmark's scope by focusing on the time span of the included terms. This approach is intended to enhance coverage and create a more thorough benchmark for new terms.
>
> > Q3: The benchmark is designed primarily for English language terms. Its applicability to other languages or multilingual models is not addressed, which could limit its relevance for researchers working on non-English or multilingual NLP systems.
>
> As mentioned in Q1 of our General Response, our benchmark is capable of generating effective test questions for new terms across different cultural regions, demonstrating its potential for broader multilingual applications. It is important to note that this approach places high demands on the capabilities of the LLMs used. Given that LLMs tend to perform poorly on some low-resource languages, the quality of the generated benchmarks may be affected. Fortunately, with the ongoing advancements in large models, there has been significant progress in multilingual capabilities, both in open-source and proprietary models. This suggests that our pipeline has the potential to create benchmarks for an increasing number of languages. We are also interested in updating the benchmark with a multilingual version in the future. Lastly, some adjustments to the prompts may be necessary to ensure the output aligns with the expected language.

---

> ### Comment · Area_Chair_nqB3 · 2024-08-29
>
> Dear reviewer, The authors have submitted a response to your review. Please ensure that you read it and respond before the end of the discussion period. If appropriate, you may update your scores based on this interaction. Even though you recommended accept, it's helpful for the authors to know whether their response addressed any of your concerns or suggestions.

---

### Author Response · Authors · 2024-08-24
**General Response (2/2)**

> Q2: How to improve the performance of LLMs upon new terms, especially how RAG-empowered LLMs perform.

The primary goal of this research is to inspire fellow scholars interested in this area to collaboratively explore solutions to the challenges posed by new terms. As an initial approach, we employed a basic RAG strategy for our preliminary tests, aiming to offer guidance and insights that can inform future enhancements.

We evaluated several different configurations. The Gold setting, as detailed in the main text, serves as an upper bound for RAG. This setting is characterized by two key assumptions: 1) it knows which term in the query is new, and 2) it reliably retrieves the exact meaning required for the question from the dictionary. In contrast, the basic RAG setting only has access to the question and does not fulfill these two criteria. Additionally, we introduced an intermediate setting, "RAG w/ term," which satisfies only the first assumption—recognizing new terms without ensuring the accuracy of the retrieved content. Our analysis focused on whether these settings offer improvements over the zero-shot Base setting.

Our experimental setup utilized the Google Search Custom Search JSON API for retrieval, consistent with the main text. In the basic "RAG" setting, we retrieved information using the question as a search query, while the "RAG w/ term" setting employed the new term itself for retrieval. We collected the top 100 results, using content from accessible web pages with valid content, or concatenating the title and snippet if the content was not usable. All retrieved information was then segmented into sentences and grouped into paragraphs of no more than 256 tokens, using Llama-2's tokenizer. For the "RAG w/ term" setting, we added a filtering step to exclude paragraphs not containing the new term.

Following the Langchain-Chatchat methodology, we embedded all paragraphs and the question using OpenAI’s text-embedding-3-large model, retrieving the 100 most relevant paragraphs based on cosine similarity. Reordering was performed using Cohere’s rerank-english-v3.0 API, selecting the top 5 ranked paragraphs as context for the model input. The results are summarized in the tables below:

| COMA 		| Base 		| RAG 		| RAG w/ term	| Gold 		|
|:--------------------------|:----------------|:-----------------|:----------------|:-----------------|
|Llama-2-Chat-7B	| 35.38		|44.59		|45.76		|54.97		|
|Llama-2-Chat-13B	| 37.57		|43.13		|43.71		|62.72		|
|Llama-2-Chat-70B	| 48.10		|57.46		|63.01		|86.11		|
|Llama-3-Inst-8B	| 54.68		|62.87		|69.01		|90.79		|
|Llama-3-Inst-70B	| 65.64		|71.78		|77.34		|96.78		|
|GPT-3.5-0125		| 54.82		|64.33		|68.86		|88.30		|
|GPT-4-1106		| 70.32		|76.46		|78.95		|97.37		|
|GPT-4-0125		| 68.86		|74.71		|78.51		|96.64		|
|Average        		| 54.42		|61.92  		|65.64  		|84.21  		|
|			|		|+7.50  	|+11.22		|+29.69	|

| COST 		| Base 	| RAG 		| RAG w/ term	| Gold 	|
|:--------------------------|:----------------|:-----------------|:----------------|:-----------------|
|Llama-2-Chat-7B	| 37.01		|55.93		|51.69		|53.25		|
|Llama-2-Chat-13B	| 43.22		|63.94		|54.24		|53.11		|
|Llama-2-Chat-70B	| 64.27		|67.51		|68.50		|75.99		|
|Llama-3-Inst-8B	| 67.94		|83.05		|81.50		|90.40		|
|Llama-3-Inst-70B	| 73.59		|81.78		|89.55		|95.62		|
|GPT-3.5-0125		| 70.06		|84.46		|79.24		|86.16		|
|GPT-4-1106		| 81.21		|92.51		|92.94		|98.16		|
|GPT-4-0125		| 79.94		|91.10		|91.24		|98.31		|
|Average        		| 64.66 		|77.54  		|76.11  		|81.38  		|
|			|		|+12.88	|+11.45		|+16.72	|

| CSJ	 		| Base 	| RAG 		| RAG w/ term	| Gold 	|
|:--------------------------|:----------------|:-----------------|:----------------|:-----------------|
|Llama-2-Chat-7B	| 83.93		|85.52		|85.52		|85.39		|
|Llama-2-Chat-13B	| 57.50		|47.94		|46.35		|61.75		|
|Llama-2-Chat-70B	| 64.67		|66.93		|62.28		|83.67		|
|Llama-3-Inst-8B	| 70.39		|81.67		|74.50		|92.16		|
|Llama-3-Inst-70B	| 64.94		|73.04		|69.46		|95.09		|
|GPT-3.5-0125		| 76.36		|61.22		|59.89		|89.24		|
|GPT-4-1106		| 77.16		|69.72		|68.92		|93.49		|
|GPT-4-0125		| 78.49		|61.22		|59.89		|94.82		|
|Average		| 71.68 		|68.41  		|65.85  		|86.95  		|
|			|		|-3.27		|-5.83		|+15.27	|

---

> ### Author Response · Authors · 2024-08-24
> **Supplement to General Response (2/2)**
>
> The efficacy of RAG varies by the specific downstream task. For COMA, the performance progression aligns with expectations: Base < RAG < RAG w/ term < Gold, showing clear performance distinctions. In contrast, for COST, naive RAG outperforms RAG w/ term, and the gap between naive RAG and Gold is minimal. Upon reviewing the retrieved content, we found that term-based searches primarily yielded definitions and Wikipedia entries. On the other hand, question-based searches retrieved sentences incorporating the term, thus providing intuitive use cases rather than mere definitions. This approach simplifies the model's task of understanding and applying new terms by leveraging a few-shot learning format to assist with fill-in-the-blank questions. Notably, weaker models like Llama-2-Chat-7B and Llama-2-Chat-13B even surpassed the performance in the Gold setting. Finally, both RAG methods showed no stable improvements in CSJ. Our observations suggest this is due to the occasionally incorrect retrieved content, as the models require the ability to discern information validity from the context. Unlike the first two tasks where correct context is assured, CSJ lacks this support, impairing the model's ability to accurately discern and utilize information from the retrieved content.
>
> Furthermore, these results confirm that our benchmark effectively evaluates the model's understanding of new terms and various enhancement strategies, comprehensively assessing the effectiveness of these methods. They also highlight the limitations of the basic RAG approach, indicating that further improvements are still needed.

---

### Author Response · Authors · 2024-08-24
**General Response (1/2)**

> Q1: Broader coverage and analysis of new terms, especially terms from different professional fields and different cultures.

This is a very interesting perspective, and we have added further analysis here. Under NewTerm 2023, we utilized GPT-4o to select technical terminology and local dialects from a pool of 300 new terms, resulting in the identification of 74 terminologies and 46 dialects. Here are some examples:

| Terminology 		| Dialect 		|
|:---------------------------------|:---------------------------------|
|front-action: adj., sense 2: In a tuba, euphonium, etc.: designating a valve placed in front of the inner tubing; (also of a musical instrument) employing such valves.| 			frontish: adj., sense 3: Trinidad and Tobago. Of a person: pushy, forward. Cf. fronting, adj. |
|dead well:n., sense 3: Chiefly Oil Industry. A depleted oil or gas well; a well from which oil or gas does not flow by its own pressure. |						straight: South African. In township slang: a full-size bottle (now 750 millilitres) of spirits. Cf. half-jack, n.|

We also assessed the accuracy of these terms compared to the average of all 300 terms from NewTerm 2023, as shown in the table below. A positive value indicates that the term is easier than average, while a negative value suggests greater difficulty. 'Base' represents zero-shot, and 'Gold' indicates the inclusion of the new terms' definitions in the prompt.

|			| Terminology 	| 		| Dialect 	| 		|
|:--------------------------|:----------------|:-----------------|:----------------|:-----------------|
|			| Base		| Gold		| Base 		| Gold		|
|Llama-2-Chat-7B	| +4.65		|+5.96		|-10.29		|-2.91		|
|Llama-2-Chat-13B	| +5.55		|+6.77		|-15.80		|-11.62		|
|Llama-2-Chat-70B	| +4.85		|+5.09		|-14.20		|-1.11		|
|Llama-3-Inst-8B	| +4.64		|+5.96		|-19.39		|+3.90		|
|Llama-3-Inst-70B	| +5.86		|+1.43		|-18.42		|+0.64		|
|Claude-Instant-1.2	| +4.36		|+1.75		|-12.52		|+0.45		|
|Claude-2.1		| +5.37		|+4.47		|-12.00		|-4.20		|
|Claude-3-haiku	| +6.51		|+1.56		|-18.90		|-1.19		|
|Claude-3-sonnet	| +7.06		|+2.81		|-23.21		|-0.22		|
|Claude-3-opus	| +8.00		|+1.95		|-14.89		|+0.42		|
|GPT-3.5-0613		| +5.26		|-0.24		|-9.36		|-0.04		|
|GPT-3.5-0125		| +4.93		|+0.94		|-11.75		|-0.94		|
|GPT-4-0613		| +5.10		|-0.04		|-13.22		|-0.24		|
|GPT-4-1106		| +4.60		|+1.20		|-16.00		|+1.58		|
|GPT-4-0125		| +6.45		|+1.54		|-17.17		|+1.67		|
|Average		| +5.55		|+2.74		|-15.14		|-0.92		|

Our findings indicate that in the Base setting, technical terminologies generally perform slightly above the average, whereas local dialects significantly underperform. This highlights the impact of cultural differences on LLM performance. Additionally, LLMs handle professional terms from general fields adeptly. However, our use of a general dictionary rather than a professional one may have limited the difficulty and specificity of the technical terminologies analyzed. Employing a more specialized dictionary could potentially alter these results. These experimental results underscore the necessity for increased focus on new terms originating from various cultures, highlighting the challenges they present to LLMs.

In the Gold setting, while the overall trend mirrors the Base setting, the magnitude of change is notably reduced. This is because when a term's definition is provided, it is presumed that the model already comprehends the term, particularly for high-performance LLMs. Consequently, the inherent difficulty of the term has a diminished impact on performance. This outcome aligns with our expectations.

---

### Decision · Program_Chairs · 2024-09-26

**Decision:**

Accept (Poster)

**Comment:**

**Overall**: This paper presents a benchmark (and a corresponding plan for continual updating of the benchmark) focused on LLM understanding of “new terms”, where “new terms” are defined as words that have recently been added to prominent online dictionaries. The benchmark allows for models with a known knowledge cutoff to be evaluated on data that is likely to be novel to them. All reviewers acknowledge the significance of the benchmark and the quality of the writing and benchmark creation. Below, I’ve summarized the main strengths and weaknesses that were brought up by reviewers. The primary weaknesses were mostly addressed in the rebuttal, though there are still some points that the authors could take into consideration in writing about the limitations of their research. However, given that none of the weaknesses challenge the importance of the dataset and methodology, I recommend that the paper be accepted on the basis of the novelty and quality of both the dataset and the procedure for ensuring it remains updated.

**Strengths**
- Writing clarity (7eHY, LL5Y)
- Multiple tasks used in assessing LLM performance (7eHY)
- The dataset creation method is mostly automated (HEp8), making it scalable and reproducible (LL5Y)
- Evolving benchmarks are an important and timely (no pun intended) contribution (HEp8, LL5Y) that presents a novel challenge (LL5Y)
- Solid methodology and analysis (LL5Y)

**Weaknesses**
- Extension to multilingual evals (or just anything other than English) is unclear (7eHY). Though the authors point out in the rebuttal that the method they introduce should be applicable more broadly, this doesn’t fully address the point raised by the reviewer because there are likely disparities between English and other languages in the availability of reputable online dictionaries that introduce new terms, and also the likelihood that all languages would be able to source equally useful new terms in this way. Careful consideration of the methodology’s applicability in each language would be needed.
- There’s the potential for bias due to the choice of online dictionaries used for sourcing the new terms (7eHY). The authors’ rebuttal does not address this point, but does an effective job of arguing that the choice of dictionaries was reasonable given the constraints of wanting an automated method. It would still be helpful for the authors to consider this as a limitation of the benchmark and discuss any potential biases, though. This also touches on concerns raised by Reviewer HEp8 about cultural implications.